# LLaMo: Large Language Model-based Molecular Graph Assistant

**Jinyoung Park**    **Minseong Bae**    **Dohwan Ko**    **Hyunwoo J. Kim**[*]
Department of Computer Science and Engineering, Korea University
{lpmn678, bms2002, ikodoh, hyunwoojkim}@korea.ac.kr

## Abstract

Large Language Models (LLMs) have demonstrated remarkable generalization and instruction-following capabilities with instruction tuning. The advancements in LLMs and instruction tuning have led to the development of Large Vision-Language Models (LVLMs). However, the competency of the LLMs and instruction tuning have been less explored in the molecular domain. Thus, we propose LLaMo: **La**rge **La**nguage Model-based **Mo**lecular graph assistant, which is an end-to-end trained large molecular graph language model. To bridge the discrepancy between the language and graph modalities, we present the multi-level graph projector that transforms graph representations into graph tokens by abstracting the output representations of each GNN layer and motif representations with the cross-attention mechanism. We also introduce machine-generated molecular graph instruction data to instruction-tune the large molecular graph-language model for general-purpose molecule and language understanding. Our extensive experiments demonstrate that LLaMo shows the best performance on diverse tasks, such as molecular description generation, property prediction, and IUPAC name prediction. The code of LLaMo is available at https://github.com/mlvlab/LLaMo.

## 1 Introduction

In recent years, molecular machine learning [1, 2, 3, 4] has received significant attention, addressing diverse tasks in the chemical domain. The predominant approach for molecular tasks is graph machine learning [5, 6, 7] that leverages the molecular graph structure, which is a natural and expressive representation of molecules. Although graph-based methods have successfully represented molecules, they have limited interpretability and incompatibility to solve multi-modal molecular tasks dealing with pairs of texts and molecules. To address these issues, recent works [4, 8] train both a language model and a graph encoder with cross-modal contrastive learning. However, the models trained with cross-modal contrastive learning are insufficient to perform open-ended molecule-to-text generation tasks [3], which are more applicable to practical use.

Large Language Models (LLMs) [2, 9, 10, 11] have shown impressive progress and accomplished human-like open-ended text generation with the power of billions of parameters. To leverage the instruction-following capability of LLMs, many works employ instruction-tuning approaches [12, 13, 14] for general-purpose language models. Motivated by the development of LLMs and instruction tuning, Large Vision-Language Models (LVLMs) have recently been explored and achieved success on image comprehension and image-to-text generation tasks [10, 11, 15, 16, 17, 18]. Despite the success of LLM-based approaches on natural language processing and machine vision domains, the research on the integration of language models and molecular graphs has been less studied due to the lack of consideration of the architecture design of Large Molecular Graph-Language Models (LMGLMs) and the molecular graph instruction data.

---

[*]corresponding author.

38th Conference on Neural Information Processing Systems (NeurIPS 2024).

In this paper, we propose LLaMo: **L**arge **La**nguage Model-based **Mo**lecular graph assistant, which seamlessly integrates a molecular graph encoder and a large language model to enable the instruction-following response generation in molecular domain. Specifically, LLaMo consists of the molecular graph encoder, large language model, and multi-level graph projector that bridges the graph encoder and large language model. The multi-level graph projector abstracts the representation of each GNN layer and motif representation using a cross-attention mechanism, ensuring a thorough understanding of molecular structures. Furthermore, we introduce machine-generated molecular graph instruction data through the pipeline to convert molecular descriptions and IUPAC names into a multi-turn conversation format. The generated instruction-following data enhances the model's ability to perform general-purpose molecule and language understanding, bridging the gap between molecular graph analysis and language-based tasks. Our proposed LLaMo outperforms the LLM-based works such as GPT-4 across diverse tasks, including molecular description generation, property prediction, and IUPAC name prediction .

Our contributions are summarized as follows:

- We propose LLaMo: **L**arge **La**nguage Model-based **Mo**lecular graph assistant consisting of graph encoder, language model, and multi-level graph projector equipped with a multi-level graph projector that captures rich information of the graph structure at multiple levels.

- We introduce GPT-4 generated molecular graph-text multi-turn conversation data to address the data scarcity problem of molecule-text datasets and improve the instruction-following capabilities of a large molecular graph-language model.

- Our experiments demonstrate that LLaMo achieves the best performance on various tasks such as molecular description generation, property prediction, and IUPAC name prediction.

## 2 Related works

**Molecular graph modeling.** Molecular graphs serve as a natural and expressive representation of molecules, effectively capturing the structural information. Graph neural networks [19, 20, 21, 22] are commonly utilized architectures for molecular graph representations. To learn graph neural networks with the limited molecular graph data [23], self-supervised learning has been explored. For example, various approaches [24, 25] have been developed to capture multi-level features of molecular graphs, such as node-level masked atom modeling [24], motif-based self-supervised learning [25, 26], and graph-level contrastive learning [27, 28]. With the advance of multi-modal large language models, molecule-language tasks such as molecule-text retrieval [8] or molecule captioning [29] have recently drawn significant attention. Recent works [3, 4, 30] have attempted to enable language models to understand molecular graphs. [30] treated nodes of molecular graphs as tokens of language models. Some works have adopted GNN-based encoders, either by propagating their outputs to language models through MLP [4] or employing cross-modal projectors [3]. However, these methods fail to consider molecular graphs at multiple levels and are hindered by inherent limitations of graph encoders, such as the over-smoothing problem [31]. To address these challenges, we propose a novel architecture, LLaMo, which effectively propagates multi-level information of molecular graphs to language models.

**Instruction tuning.** Recent advancements of LLMs lead to extensive research on *instruction tuning*, aimed at improving the model's capability to follow human instructions [12, 32, 33, 34, 35]. To construct high-quality instruction tuning data, a line of previous approaches [34, 35] has adopted existing human-annotated datasets and integrated them with a new structure and template. On the other hand, recent studies [12, 36, 37] on instruction tuning have collected data samples from strong LLMs like GPT-4 [10]. These works first manually construct annotated seed instruction samples and expand them by prompting LLMs. As a result, several instruction-tuned LLMs [14, 16, 37] have been proposed from the open-source LLMs, *e.g.*, LLaMA [9] and shown generalizability across a wide range of instructions. More recently, those studies on instruction tuning have been expanded to visual instruction tuning in image [15, 17, 38] and video [39, 40] domain to enable the model to understand the visual contents. Inspired by the instruction tuning for multi-modal LLMs in other domains, in this work, we study instruction tuning specifically for molecule graphs, which has been underexplored in the literature.

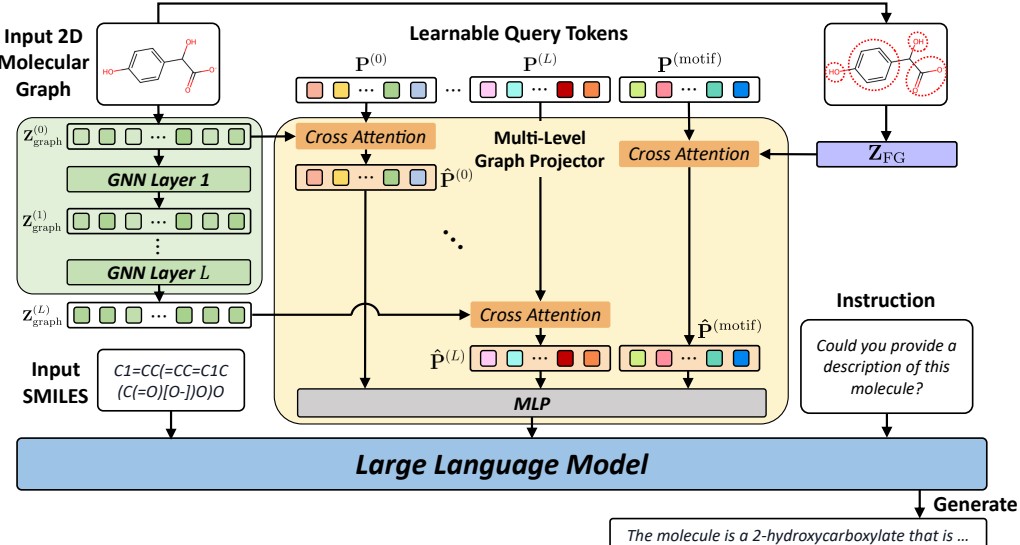

Figure 1: Overall framework of LLaMo. LLaMo consists of a graph neural network, a multi-level graph projector, and a large language model. It first encodes an input 2D molecular graph with the graph neural network and then converts the encoded graph into molecular graph tokens with the multi-level graph projector. Finally, the large language model generates the instruction-following response given the input SMILES, graph tokens, and the instruction.

## 3 LLaMo: Large Language Model-based Molecular Graph Assistant

The primary goal is to seamlessly integrate a molecular graph encoder and a Large Language Model (LLM) to generate instruction-following responses to the input texts and molecules. To achieve it, we propose LLaMo: **La**rge **La**nguage Model-based **Mo**lecular graph assistant, a general-purpose Large Molecular Graph-Language Model (LMGLM) equipped with a multi-level graph projector. Specifically, the proposed framework utilizes three input modalities: 1D SMILES [41], 2D molecular graph, and text (instruction). SMILES [41] is a 1D representation of a molecule, and a 2D molecular graph is processed by a GNN. The three input modalities are fed as a sequence of tokens and our LLaMo autoregressively generates text responses. Formally, given SMILES $\mathbf{S}$, molecular graph tokens $\mathbf{G}$, and text (instruction) $\mathbf{T}$, the proposed method renders the response $\mathbf{Y} = \{\mathbf{y}_i\}_{i=1}^{K}$ as:

$$p\left(\mathbf{Y}|\mathbf{S}, \mathbf{G}, \mathbf{T}\right) = \prod_{i=1}^{K} p\left(\mathbf{y}_i|\mathbf{S}, \mathbf{G}, \mathbf{T}, \mathbf{y}_{<i}\right), \tag{1}$$

where $\mathbf{y}_{<i}$ indicates generated token sequences until $i$-th token.

### 3.1 Model Architecture

The overall architecture of LLaMo is illustrated in Figure 1. LLaMo consists of a graph encoder, a multi-level graph projector, and a backbone large language model. The graph encoder $g(\cdot)$ takes a 2D molecule graph as an input and outputs their node representations as a sequence of tokens. The multi-level graph projector $\text{Proj}_{\text{MG}}(\cdot)$ transforms the sequence of node representations into molecular tokens to align them with the LLM. Then, the LLM $f(\cdot)$ processes molecular and text tokens and provides a response in an autoregressive manner.

**Graph encoder.** We adopt Graph Neural Networks (GNNs) as a molecular graph encoder. Given the graph $\mathcal{G}$, graph neural networks $g(\cdot)$ iteratively update node representation $\mathbf{z}_v^{(l)} \in \mathbb{R}^{d^{(l)}}$ via the message-passing framework. With the message-passing, $L$-layer GNN provides node representations $\mathbf{z}_v^{(L)}$ that express an $L$-hop ego-graph given the node $v$ as a center node. More details about graph neural networks are in the Appendix C.

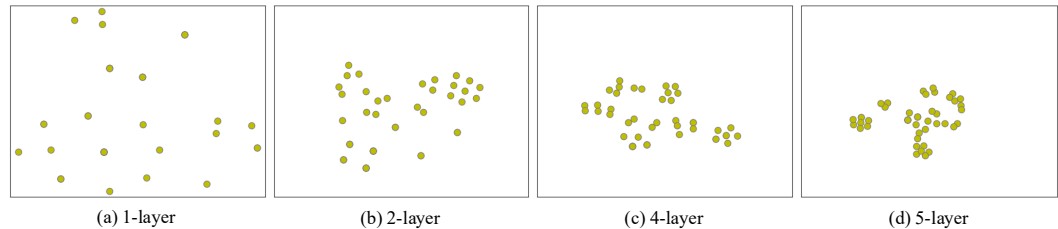

| (a) 1-layer | (b) 2-layer | (c) 4-layer | (d) 5-layer |

Figure 2: Node representations of graph encoder with 1,2,4,5 layers. As the number of layers increases, node representations collapse.

**Multi-level graph projector.** The goal of a multi-level graph projector is to align the graph encoder with the LLM by transforming a set of node representations $\mathbf{Z}_{\text{graph}}^{(L)}$ into a sequence of molecular graph tokens $\mathbf{H}_{\text{graph}}$. It enables the language model to utilize graph information. In the literature, projectors have been proposed mainly for Large Vision-Language Models (LVLMs) [17, 18, 38, 42, 43]. They are usually implemented using a linear projection [38] or an abstraction of visual features [17, 42], which are outputs of the final layer of a visual encoder given input image. Analogously, we can design the projector for large molecular graph-language models with a linear projection or an abstraction of high-level node representations from the pre-trained graph encoder, which is formulated as:

$$\mathbf{H}_{\text{graph}} = \text{Proj}\left(\mathbf{Z}_{\text{graph}}^{(L)}\right), \text{ where } \mathbf{Z}_{\text{graph}}^{(L)} = g\left(\mathcal{G}\right), \tag{2}$$

where $\mathbf{Z}_{\text{graph}}^{(L)} = \left[\mathbf{z}_0^{(L)}, \ldots, \mathbf{z}_{|\mathcal{V}|}^{(L)}\right] \in \mathbb{R}^{|\mathcal{V}| \times d^{(L)}}$ is the concatenation of node representation $\mathbf{z}_v^{(L)} \in \mathbb{R}^{d^{(L)}}$ from $L$-th layer GNN and $\text{Proj}\left(\cdot\right)$ is the projector.

However, we observe that the high-level representation is not effective in capturing the local information due to the over-smoothing problem [31], which means that the node representations become indistinguishable, as the number of layers in the GNN increases. Figure 2 depicts node representations (yellow dots) of graph encoder with 1,2,4,5 layers on one molecular graph sample. (More samples are in Appendix I.) As mentioned above, node representations become over-smoothed as the number of layers increases, leading to nearly identical node representations in the final layer. Consequently, conventional projectors relying on high-level node representations have a limited capability to preserve the detailed or local information of molecular graphs. Moreover, many tasks require multi-scale information, including atom, atomic group, and molecule levels. Hence, the projector that solely utilizes features from the top layer is suboptimal for the tasks.

Motivated by the observations, we propose a novel multi-level graph projector to generate graph tokens that contain richer information reflecting the graph structure at multiple levels. The multi-level graph projector $\text{Proj}_{\text{MG}}\left(\cdot\right)$ is formulated as

$$\mathbf{H}_{\text{graph}} = \text{Proj}_{\text{MG}}\left(\left\{\mathbf{Z}_{\text{graph}}^{(l)}\right\}_{l=0}^{L}\right), \text{ where } \left\{\mathbf{Z}_{\text{graph}}^{(l)}\right\}_{l=0}^{L} = g\left(\mathcal{G}\right). \tag{3}$$

The method captures multi-hop graph information by leveraging node representations from all layers of a GNN. To handle an arbitrary number of nodes, yielding a variable length $|\mathcal{V}| \times L$ features, we adopt the cross-attention with learnable tokens $\mathbf{P}^{(l)} = \left[\mathbf{p}_1^{(l)}, \ldots, \mathbf{p}_b^{(l)}\right] \in \mathbb{R}^{b \times d}$ for $l = 0, \ldots, L$, where $b$ is the number of learnable prompts. Here, $[\cdot, \cdot]$ indicates the concatenation operation. The learnable tokens aggregate $l$-th layer GNN representations into a fixed number of tokens as:

$$\hat{\mathbf{P}}^{(l)} = \text{Attn}^{(l)}\left(\mathbf{P}^{(l)}, \mathbf{Z}_{\text{graph}}^{(l)}, \mathbf{Z}_{\text{graph}}^{(l)}\right) \in \mathbb{R}^{b \times d}, \tag{4}$$

where $\text{Attn}\left(Q, K, V\right)$ is the attention operation with query $Q$, key $K$, and value $V$.

For more detailed representations of the input molecule, LLaMo also has learnable tokens $\mathbf{P}^{(\text{motif})}$ for motif-level representations. We use the functional groups as motifs, which are the statistically important subgraphs in the molecular graphs. To construct functional group representations $\mathbf{Z}_{\text{FG}}$, we initially identify functional groups, following [23]. Then, we vectorize the main characteristics of each functional group, which is represented as $\mathbf{z}_{\text{FG},i}$. Finally, the functional group representations $\mathbf{Z}_{\text{FG}}$

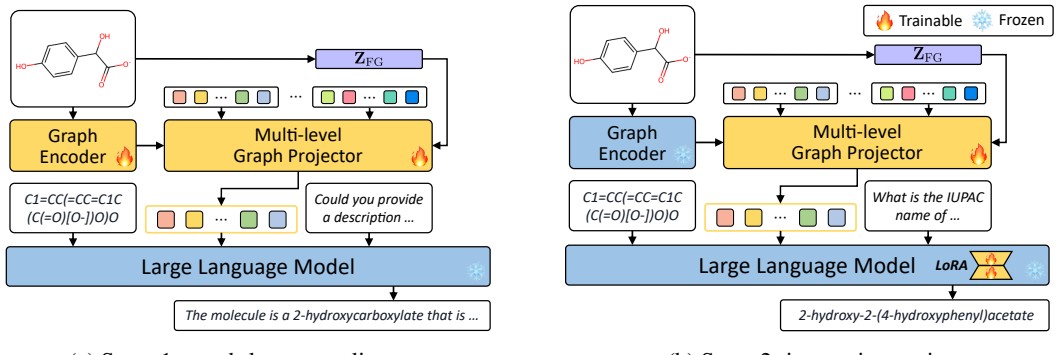

(a) Stage 1: graph-language alignment      (b) Stage 2: instruction-tuning

Figure 3: Two-stage training pipeline. Stage 1 involves training the graph encoder, and stage 2 entails fine-tuning the LLM using LoRA. In both stages, the multi-level graph projector is continuously trained. All training processes are performed by generating the instruction-following response.

are constructed by concatenating all individual functional group representations $\mathbf{z}_{\text{FG},i}$, which is formulated as $\mathbf{Z}_{\text{FG}} = [\mathbf{z}_{\text{FG},0}, \ldots, \mathbf{z}_{\text{FG},M}]$, where $M$ indicates the number of the functional groups in the given molecular graph. Given the functional group representations $\mathbf{Z}_{\text{FG}}$ of the input molecule, we obtain $\hat{\mathbf{P}}^{(\text{motif})} = \text{Attn}^{(\text{motif})}\left(\mathbf{P}^{(\text{motif})}, \mathbf{Z}_{\text{FG}}, \mathbf{Z}_{\text{FG}}\right)$ with the cross-attention.

Then, we obtain the graph-level representations by applying MLP to the multi-hop and motif-level representations, *i.e.*, $[\hat{\mathbf{P}}^0, \ldots, \hat{\mathbf{P}}^{(L)}, \hat{\mathbf{P}}^{(\text{motif})}]$. It is formulated as:

$$\mathbf{H}_{\text{graph}} = \text{MLP}\left(\left[\hat{\mathbf{P}}^{(0)}, \ldots, \hat{\mathbf{P}}^{(L)}, \hat{\mathbf{P}}^{(\text{motif})}\right]\right) \in \mathbb{R}^{b \cdot (L+2) \times d}, \tag{5}$$

where $\mathbf{H}_{\text{graph}}$ is a sequence of graph tokens to be fed into the LLM.

**Large language models.** After constructing tokens for encoding molecular graphs, LLaMo fuses SMILES representation, graph, and text tokens and puts them into the language model to generate an instruction-following response.

### 3.2 Training LLaMo

Similar to most LVLMs [18, 38, 42], we train LLaMo in the two-stage pipeline: (1) pre-training for molecular graph-language alignment, and (2) instruction-tuning end-to-end as in Figure 3.

**Stage 1. Pre-training for molecular graph-language alignment.** The first stage focuses on the alignment between the graph encoder and a large language model by learning our multi-level graph projector. In this stage, with the LLM frozen, we train the multi-level graph projector and the graph encoder by generating molecule descriptions. For training, we use a molecule-description pair dataset (*e.g.,* PubChem [44]) consisting of a 1D SMILES representation of molecule and molecular graph and its corresponding description.

**Stage 2. Instruction-tuning end-to-end.** In the second stage, we train the LLM to enhance the instruction-following capabilities and enable a deeper understanding of molecular graphs. In this stage, we freeze the graph encoder and train both the multi-level graph projector and the LLM. Since it is too expensive to train the full LLM, we employ LoRA [45] to adapt LLM to the data. For instruction-following, we use the GPT-generated instruction-following multi-turn conversation dataset, which will be introduced in Section 4. In addition to our generated instruction-following dataset, we use a diverse set of datasets with various instructions: molecule description generation, molecular property prediction, IUPAC name generation, forward reaction prediction, and retrosynthesis datasets.

## 4 GPT-assisted Molecular Graph Instruction Data Generation

Instruction data are essential for improving the instruction-following capabilities of LLM-based models. Despite active research on instruction-tuning, the instruction-following data for molecular

Table 1: **One example to illustrate the instruction-following data.** The top block shows the contexts such as SMILES, description, and IUPAC name used to prompt GPT, and the bottom block shows the response of GPT-4.

---

**Context type 1: SMILES representation**
CCCCC(C)/C=C(\C)/C=C/C(=O)NC1=C[C@]([C@@H](CC1=O)O)(/C=C/C=C/C=C/C(=O)NC2=C(CCC2=O)O)O
**Context type 2: Description**
The molecule is a polyene antibiotic that is TMC-1A in which the 2,4-dimethyloct-2-enoyl group has been replaced by an (E,E)-4,6-dimethyldeca-2,4-dienoyl group. TMC-1C is an antitumour antibiotic isolated from Streptomyces sp. A-230...<omitted>
**Context type 3: IUPAC name**
(2E,4E)-N-[(3S,4R)-3,4-dihydroxy-3-[(1E,3E,5E)-7-[(2-hydroxy-5-oxocyclopenten-1-yl)amino]-7-oxohepta-1,3,5-trienyl]-6-oxocyclohexen-1-yl]-4,6-dimethyldeca-2,4-dienamide

---

**Response: Multi-turn conversation**
Question:
What is the IUPAC name of the molecule you are analyzing?
===
Answer:
The IUPAC name of the molecule is (2E,4E)-N-[(3S,4R)-3,4-dihydroxy-3-[(1E,3E,5E)-7-[(2-hydroxy-5-oxocyclopenten-1-yl)amino]-7-oxohepta-1,3,5-trienyl]-6-oxocyclohexen-1-yl]-4,6-dimethyldeca-2,4-dienamide.
...
===
Question:
Can you identify the type of acid or base this molecule can act as in a reaction?
===
Answer:
This molecule can act as a weak acid due to the presence of the carboxylic acid group.

---

graphs have been less explored in the literature since annotations require expertise. To alleviate the need for expertise and minimize the manual efforts, we utilize GPT-4 [10] to generate molecular graph-text instruction-following data using graph-text pair datasets.

Inspired by previous results [46, 47], we construct multi-turn conversation datasets, which are more diverse and effective for instruction tuning compared to simple pairs of questions and answers. We leverage GPT-4 to generate multi-turn conversations with tailored contexts/prompts that consist of two representations for molecular graphs and description: (i) SMILES representation that describes the chemical structures with special strings, (ii) captions that explain the molecule, and (iii) IUPAC name that describes the molecule based on its chemical composition and structure. These representations enable the GPT-4, which inherently lacks in-depth molecular knowledge, to understand and generate a diverse and high-quality set of examples. One example of the input representations is shown in the top block of Table 1.

Specifically, we generate the multi-turn conversation data in three steps: **1)** select exemplar conversations among machine-generated instruction-tuning data, **2)** generate multi-turn conversations via in-context learning with the exemplar conversations as prompts, and **3)** filter out incomplete conversations and those with many turns. In the first step, we generate exemplars with a brief human-written instruction as shown in Appendix H. However, we found that GPT-4 frequently fails to generate complete multi-turn conversations without the exemplars. To address this issue, we generate the instruction data with in-context learning. We sample exemplars from a small set of complete conversations generated by GPT-4 in the first step. Then, GPT-4 generates the complete multi-turn conversation data for the instruction tuning guided by the prompts wrapped with the generated exemplars. To validate the quality of the generated conversation, we sample 500 subsets generated via in-context learning. We find that some conversations consisting of a large number of turns are prone to generating incomplete and inaccurate outputs. So, we filter out incomplete conversations and those with many turns. The example of the generated multi-turn conversation is in the bottom block of Table 1. In total, we generate 12K unique molecular graph-language instruction-following samples using PubChem324k dataset [3, 44].

Table 2: Performance (%) of generalist models on three tasks: molecule description generation, IUPAC prediction, and property prediction. **Mol. Inst. tuned** denotes the molecular instruction-tuned model. ∗ The result is not available since LLaMA2 fails generating numerical outputs. † denotes the experimental results drawn from Mol-Instruction [48].

| Model | LLM | Mol. Inst. tuned | Molecule Description | | IUPAC Prediction | | Property pred. |
| | | | BLEU (↑) | METEOR (↑) | BLEU (↑) | METEOR (↑) | MAE (↓) |
|---|---|---|---|---|---|---|---|
| GPT-3.5 | GPT-3.5 | | 2.2 | 19.7 | 33.4 | 52.6 | 0.075 |
| GPT-3.5 (ICL) | GPT-3.5 | | 28.4 | 56.1 | 50.3 | 62.0 | 0.028 |
| GPT-4 | GPT-4 | | 0.8 | 16.7 | 29.0 | 48.1 | 0.098 |
| GPT-4 (ICL) | GPT-4 | | 27.0 | 52.2 | 51.8 | 62.4 | 0.019 |
| Galactica† | Galactica | | 0.8 | 6.5 | – | – | 0.568 |
| Text+Chem T5† | T5-Base | | 3.6 | 13.9 | – | – | – |
| LLaMA2 | LLaMA2-7B | | 0.0 | 14.1 | 0.0 | 0.4 | N/A∗ |
| Mol-Instructions† | LLaMA2-7B | ✓ | 14.3 | 25.4 | – | – | 0.013 |
| **LLaMo (Ours)** | LLaMA2-7B | ✓ | **38.9** | **67.1** | **56.0** | **73.2** | **0.006** |

# 5 Experiments

## 5.1 Experimental Settings

**Benchmarks.** To evaluate the efficacy of the proposed method, we evaluate the model for three tasks such as **1)** molecule description generation, **2)** IUPAC name prediction, **3)** property prediction (regression). We conducted experiments under two major settings: generalist and specialist models. In the generalist setting, one model handles all three tasks, whereas in the specialist setting, we train a model for each downstream task. More details about benchmarks are in Appendix G.

**Implementation details.** For the generalist models, we train our LLaMo based on `Llama-2-7b-chat` [9] for a fair comparison with Mol-Instructions [48]. For the specialist models, we train our LLaMo with Galactica 1.3B [2] for a fair comparison with MolCA [3]. To train the generalist variant of LLaMo, we use a training split of molecular description generation dataset of Mol-Instruction [48] in stage 1. In stage 2, the model is instruction-tuned with a training split of description generation, property prediction, forward reaction, and retrosynthesis instruction dataset of Mol-Instruction [48], IUPAC name prediction from [3], and our GPT-generated instruction-following data. To train the specialist variant of LLaMo, we follow MolCA [3] to train the model with a pretraining split of PubChem324kV2 in the stage 1 phase and fine-tune the model for each specific downstream task in the stage 2. We adopt a long training schedule (epoch 1 pre-training, epoch 3 instruction tuning) for the final models. For analysis, we use a short training schedule (epoch 1 pre-training, epoch 1 instruction tuning). For further implementation details, refer to Appendix E.1.

**Baselines.** For the generalist models, we compare our LLaMo with (1) LLM-based generalist models including Galactica [2], LLaMA2-7B [9], GPT-3.5, and GPT-4, (2) Molecule-specialized LLM such as Text+Chem T5 [49], and (3) Molecule instruction-tuned generalist model such as Mol-Instructions [48]. Since GPT-3.5 and GPT-4 have difficulty in solving the tasks without in-context learning, we additionally measure the performance of GPT-3.5 and GPT-4 with 4-shot in-context learning, which are GPT-3.5 (ICL) and GPT-4 (ICL). For the specialist models, we use single-task specialist molecule-language models as baselines, including MolT5 [29], MoMu [4], and MolCA [3].

## 5.2 Experimental Results

**Generalist models.** We provide the experimental results of generalist models in molecular description generation, IUPAC name generation, and property prediction tasks. Our LLaMo is built on LLaMA-7B and it is fine-tuned by our instruction-tuning method. Table 2 shows that our LLaMo achieves the best performance in all three tasks. In comparison to **GPT-4 (ICL)**, which is GPT-4 with in-context-learning, LLaMo shows a performance improvement of 11.9 in BLEU-4 and 14.9 in METEOR for molecular description generation. Furthermore, LLaMo outperforms Mol-Instructions, an instruction-tuned model with molecular data, by a substantial performance gain of 41.7 in METEOR for molecular description generation and a 0.007 performance gain in MAE on the property prediction task. More experimental results on forward reaction prediction and retrosynthesis are in Appendix D.

Table 3: Performance (%) of specialist models on molecule captioning with the PubChem324k and ChEBI-20 datasets and IUPAC name prediction. Full ft denotes full parameter fine-tuning.

| Model | LLM | Training type | PubChem324kV2 | | ChEBI-20 | | IUPAC |
| | | | BLEU | METEOR | BLEU | METEOR | METEOR |
|---|---|---|---|---|---|---|---|
| MolT5-Small | T5-Small | full ft | 8.5 | 18.5 | 43.6 | 55.1 | 42.5 |
| MolT5-Base | T5-Base | full ft | 20.9 | 35.6 | 45.7 | 56.9 | 53.2 |
| MolT5-Large | T5-Large | full ft | 22.2 | 36.6 | 50.8 | 61.4 | 58.5 |
| MoMu-Small | T5-Small | full ft | 12.0 | 21.8 | 44.5 | 57.6 | – |
| MoMu-Base | T5-Base | full ft | 21.5 | 34.2 | 46.2 | 57.6 | – |
| MoMu-Large | T5-Large | full ft | 22.8 | 36.2 | 51.5 | 59.7 | – |
| MolCA, Galac$_{125M}$ | Galactica-125M | full ft | 24.3 | 41.6 | 52.6 | 63.6 | 71.8 |
| MolCA, Galac$_{1.3B}$ | Galactica-1.3B | LoRA | 30.3 | 45.6 | 53.1 | 65.1 | 72.1 |
| **LLaMo (Ours)** | Galactica-1.3B | LoRA | **34.4** | **48.0** | **54.8** | **66.6** | **73.4** |

Table 4: Performance comparison according to the projector type.

| Projector | Molecule description | | IUPAC prediction | | Property QA |
| | BLEU ($\uparrow$) | METEOR ($\uparrow$) | BLEU ($\uparrow$) | METEOR ($\uparrow$) | MAE ($\downarrow$) |
|---|---|---|---|---|---|
| w/o Graph | 26.1 | 56.6 | 36.3 | 62.2 | 0.013 |
| MLP (w/ low-level) | 32.4 | 62.1 | 42.2 | 68.4 | 0.009 |
| MLP (w/ high-level) | 33.8 | 63.4 | 45.5 | 67.4 | 0.008 |
| MLP (w/ concat) | 34.8 | 64.1 | 47.1 | 70.2 | **0.007** |
| Resampler | 34.4 | 62.8 | 43.4 | 65.2 | 0.009 |
| MGProj (w/o motif) | 36.1 | 65.3 | 48.8 | 69.8 | 0.008 |
| **MGProj (Ours)** | **37.8** | **66.1** | **49.6** | **70.9** | **0.007** |

**Specialist models.** We also evaluate the performance of specialist models to validate the effectiveness of our LLaMo, which is individually fine-tuned for each dataset. Table 3 demonstrates that our LLaMo consistently achieves the best performance across all tasks and datasets. Specifically, LLaMo outperforms the second-best model MolCA with Galactica 1.3B, by 4.1 in BLEU-score and 2.4 in METEOR on the PubChem324kV2 dataset. For IUPAC name prediction, LLaMo also shows superior performance, achieving a METEOR score of 73.4, which surpasses MolCA with Galactica 1.3B by a margin of 1.3 points. This experimental result indicates that our LLaMo is consistently effective in comprehending molecular graphs based on diverse large language models.

### 5.3 Analysis

**Impact of multi-level graph projector.** To validate the effectiveness of our multi-level graph projector, we compare the performance of the multi-level graph projectors (denoted by **MGProj**) with other projectors in Table 4, including two widely-used projectors such as MLPs and resamplers. Additionally, we measure the performance of the base model without a graph (and a projector) denoted as **w/o Graph** for the ablation study. **MLP (w/ low-level)** and **MLP (w/ high-level)** denote the MLP projectors where the input is low-level representation $\mathbf{Z}_{\text{graph}}^{(1)}$ and high-level representation $\mathbf{Z}_{\text{graph}}^{(L)}$, respectively. **MLP (w/ concat)** indicates the MLP projector with the concatenated representations of all GNN layers as an input. **Resampler** denotes the cross-attention based resampler projector designed in Qwen-VL [50]. **MGProj (w/o motif)** and **MGProj** are our multi-level graph projector without and with motif tokens $\hat{\mathbf{P}}^{(\text{motif})}$.

Table 4 shows that our multi-level graph projector (**MGProj**) achieves the best performance across all three tasks. Specifically, the multi-level graph projector achieves 49.6 BLEU and 70.9 METEOR scores with a significant improvement compared to MLP projectors in the IUPAC prediction task. These experimental results demonstrate that our multi-level graph projector is more effective than conventional projectors by capturing multi-scale information, including atom, atomic group, and molecule-level information.

Table 5: Ablation studies on training stage and GPT-generated instruction tuning data.

| Stage 1 | Stage 2 | GPT-generated data | Molecule description | | IUPAC prediction | | Property QA |
|---|---|---|---|---|---|---|---|
| | | | BLEU (↑) | METEOR (↑) | BLEU (↑) | METEOR (↑) | MAE (↓) |
| | | | 0.0 | 14.1 | 0.0 | 0.4 | N/A |
| ✓ | | | 35.5 | 64.8 | 7.3 | 16.9 | N/A |
| ✓ | ✓ | | 37.2 | 65.1 | 47.5 | 70.2 | **0.007** |
| ✓ | ✓ | ✓ | **37.8** | **66.1** | **49.6** | **70.9** | **0.007** |

Table 6: Performance comparison according to the training type.

| Training type | Molecule description | | IUPAC prediction | | Property QA |
|---|---|---|---|---|---|
| | BLEU (↑) | METEOR (↑) | BLEU (↑) | METEOR (↑) | MAE (↓) |
| w/o inst. tuning (Stage 1) | 35.5 | 64.8 | 7.3 | 16.9 | N/A |
| Multi-task | 36.9 | 64.2 | 49.4 | 70.5 | 0.218 |
| **Instruction-tuning (Ours)** | **37.8** | **66.1** | **49.6** | **70.9** | **0.007** |

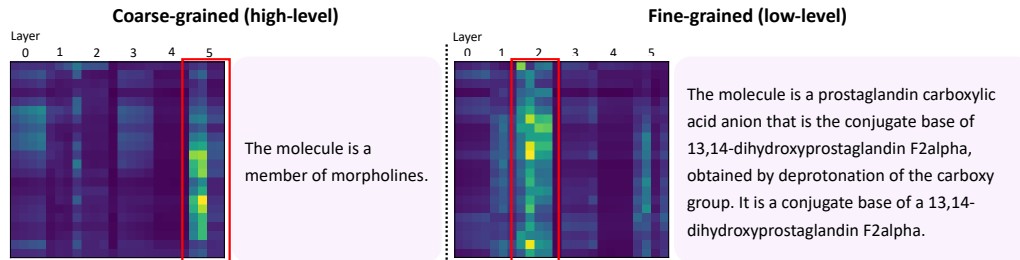

Figure 4: Visualization of attention maps for samples with coarse-grained caption (left) and fine-grained caption (right). The attention scores of high-level features are relatively high when generating coarse-grained captions, whereas those of low-level features are high for fine-grained captions.

**Impact of GPT-generated instruction-tuning data.** In Table 5, we provide the ablation studies of each training stage and our GPT-generated instruction dataset. The experimental results reveal that the instruction tuning with our generated multi-turn conversation data enhances the performance of LLaMo compared to the models trained via one or two-stage training without our GPT-generated instruction data. This indicates that instruction tuning with our GPT-generated multi-turn conversation data provides the model with more detailed and instruction-following guidance.

**Instruction tuning v.s. multi-task learning.** Table 6 shows the advantages of instruction-tuning based on task instructions compared to multi-task learning using the simple task identifier. We use the task name as a simple task identifier for multi-task learning. From the table, the model without instruction tuning (Stage 1) achieves BLUE score of 35.5 and 7.3 on molecule description and IUPAC prediction tasks, respectively. The multi-task learning approach improves the scores to 36.9 for molecule description and 49.4 for IUPAC prediction. However, the instruction-tuning method demonstrated the most significant enhancement, achieving the highest scores of 37.8 for molecule description and 49.6 for IUPAC prediction. These results indicate that instruction tuning outperforms both the baseline and multi-task learning methods, suggesting its effectiveness in improving model performance on general-purpose training.

**Visualization of attention maps.** We visualize the attention map to explore the effect of the multi-level graph projector in Figure 4. The figure illustrates the attention maps of graph tokens for generating coarse-grained (left) and fine-grained (right) descriptions. Interestingly, the attention scores of the low-level are relatively higher than the high-level when generating fine-grained captions, whereas the attention value of the high levels is high when generating coarse-grained captions. This indicates that both low and high-level graph structural information is crucial in expressing the molecules, and the attention matrix is adaptive to the caption types.

**Qualitative analysis.** Figure 5 shows a GT description and the molecular descriptions generated by the model with and without the molecular graph (SMILES representation only). As shown in the

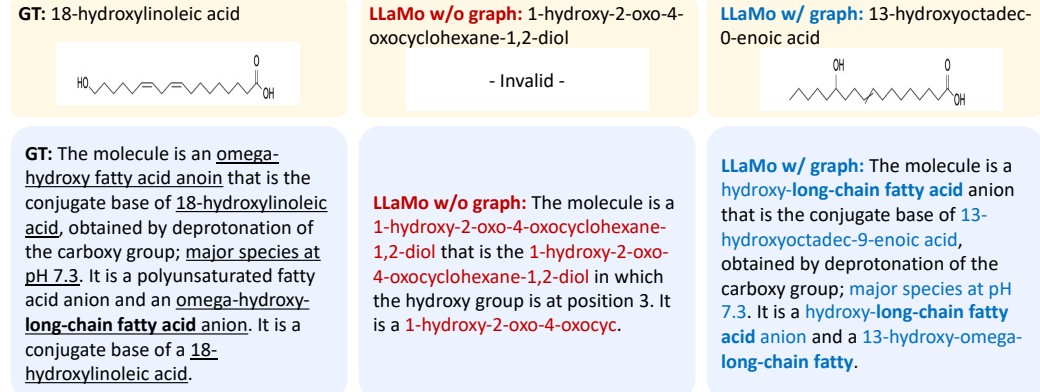

**GT:** 18-hydroxylinoleic acid

**LLaMo w/o graph:** 1-hydroxy-2-oxo-4-oxocyclohexane-1,2-diol

- Invalid -

**LLaMo w/ graph:** 13-hydroxyoctadec-0-enoic acid

**GT:** The molecule is an omega-hydroxy fatty acid anoin that is the conjugate base of 18-hydroxylinoleic acid, obtained by deprotonation of the carboxy group; major species at pH 7.3. It is a polyunsaturated fatty acid anion and an omega-hydroxy-long-chain fatty acid anion. It is a conjugate base of a 18-hydroxylinoleic acid.

**LLaMo w/o graph:** The molecule is a 1-hydroxy-2-oxo-4-oxocyclohexane-1,2-diol that is the 1-hydroxy-2-oxo-4-oxocyclohexane-1,2-diol in which the hydroxy group is at position 3. It is a 1-hydroxy-2-oxo-4-oxocyc.

**LLaMo w/ graph:** The molecule is a hydroxy-long-chain fatty acid anion that is the conjugate base of 13-hydroxyoctadec-9-enoic acid, obtained by deprotonation of the carboxy group; major species at pH 7.3. It is a hydroxy-long-chain fatty acid anion and a 13-hydroxy-omega-long-chain fatty.

Figure 5: An example of molecular description generation results of LLaMo w/o graph and LLaMo w/ graph given the molecule ("C(CCC/C=C\C/C=C\CCCCCO)CCCC(=O)[O-1]"). In the top box, the molecular graphs of IUPAC and functional groups in the descriptions are depicted.

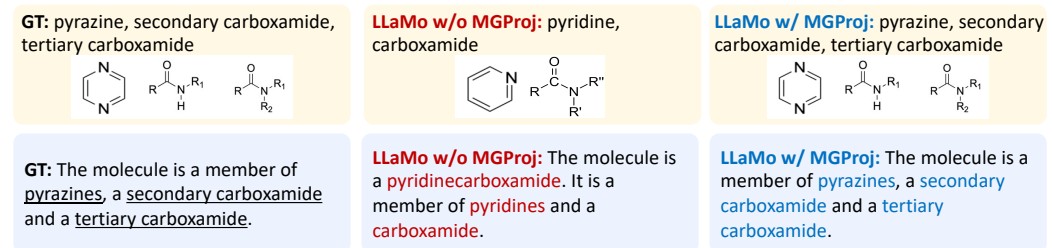

**GT:** pyrazine, secondary carboxamide, tertiary carboxamide

**LLaMo w/o MGProj:** pyridine, carboxamide

**LLaMo w/ MGProj:** pyrazine, secondary carboxamide, tertiary carboxamide

**GT:** The molecule is a member of pyrazines, a secondary carboxamide and a tertiary carboxamide.

**LLaMo w/o MGProj:** The molecule is a pyridinecarboxamide. It is a member of pyridines and a carboxamide.

**LLaMo w/ MGProj:** The molecule is a member of pyrazines, a secondary carboxamide and a tertiary carboxamide.

Figure 6: An example of molecular description generation results of LLaMo w/o MGProj and LLaMo w/ MGProj given the molecule ("C[C@@H]1CN(C(=O)C2=C(C(=CC=C2)NC(=O)C3=NC=CN=C3)O[C@@H]1CNC)[C@H](C)CO"). In the top box, the molecular graphs of IUPAC and functional groups in the descriptions are depicted.

figure, LLaMo with a graph denoted as **LLaMo w/ graph** generates a better molecular description compared to LLaMo without a graph (**LLaMo w/o graph**). The GT description explains the molecule with 'omega-hydroxy-long-chain fatty acid anion'. Since LLaMo w/o graph does not have any graph structural information, it fails to generate a description with an invalid IUPAC name ('1-hydroxy-2-oxo-4-oxocyclohexane-1,2-diol'), while LLaMo w/ graph generates a more related description with 'hydroxy-long-chain fatty acid anion'. In addition, we know that LLaMo w/ graph accurately predicts the long-chain structure of the molecule.

We also perform another qualitative analysis by comparing molecular descriptions generated from the model with and without our Multi-level Graph Projector (MGProj) denoted by **LLaMo w/ MG Proj** and **LLaMo w/o MGProj** in Figure 6. The figure shows that the multi-level graph projector plays a crucial role in capturing the details of the molecule. Compared to **LLaMo w/o MGProj** generating 'pyridine', the model with MGProj generates accurate molecular description including 'pyrazine' same as GT description. This demonstrates that the multi-level graph projector is effective in molecule understanding and generation by preserving the molecular graph structural information.

## 6   Conclusion

We propose LLaMo: Large Language Model-based Molecular graph assitant, an end-to-end trained Large Molecular Graph Language Model, to perform various molecule-related tasks with a single model. For the projector, we newly introduce a multi-level graph projector, which addresses the over-smoothing problem of the graph encoder and captures multi-hop graph information. We also present machine-generated instruction-following data in the form of multi-turn conversations to improve the instruction-following capabilities of the large language model.

## Acknowledgement

This work was partly supported by ICT Creative Consilience Program through the Institute of Information & Communications Technology Planning & Evaluation (IITP) (IITP-2024-RS-2020-II201819, 10%) and the National Research Foundation of Korea (NRF) (NRF-2023R1A2C2005373, 45%) grant funded by the Korea government (MSIT), and the Virtual Engineering Platform Project (Grant No. P0022336, 45%), funded by the Ministry of Trade, Industry & Energy (MoTIE, South Korea). We appreciate Dr. Jaesung Kwak for valuable comments and discussions.

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

The appendix is organized into the following sections.

## A  Limitations

Our LLaMo is built upon an LLM, *e.g.*, LLaMA and Galactica, and is fine-tuned on molecule benchmark datasets by leveraging its pretrained knowledge. Given that LLMs are pretrained using extensive web-crawled corpora, it is uncertain whether the data used for LLMs' pretraining and the test samples in molecule benchmark datasets are mutually exclusive. This results in implicit data leakage when fine-tuning and evaluating LLaMo on molecule benchmark datasets. Furthermore, LLMs inherently require large memory and computational costs and cause hallucination problems where the model generates incorrect but plausible text. Our LLaMo may inherit these LLMs' problems due to LLMs' powerful pre-trained knowledge.

## B  Broader Impacts

We proposed the first molecular graph-based general-purpose model, LLaMo, which is widely applicable to various molecule tasks such as molecule captioning, property prediction, and IUPAC naming. Our LLaMo itself does not have negative societal impacts. However, as discussed above, since our model is built upon an LLM, the model sometimes generates biased output concerning race, religion, culture, and gender, resulting in the misusage of our model. Also, training LLMs requires massive amounts of $CO_2$ emission promoting global warming.

## C  Explanation on Graph Neural Networks

Let $\mathcal{G} = (\mathcal{V}, \mathcal{E}, \mathbf{X})$ denote the input graph, where $\mathcal{V}, \mathcal{E}$ are a set of nodes and edges, respectively, and $\mathbf{X}$ indicates a set of input node features. The input feature of node $v \in \mathcal{V}$ is defined as $\mathbf{x}_v$ and the edge between node $u$ and $v$ is represented with $(u, v) \in \mathcal{V} \times \mathcal{V}$. The neighbor set of node $v$ on the input graph is denoted by $\mathcal{N}_v = \{u | (u, v) \in \mathcal{E}\}$. Given the graph, graph neural networks iteratively update node representation $\mathbf{z}_v^{(l)} \in \mathbb{R}^{d^{(l)}}$ via the following message-passing framework:

$$\mathbf{z}_v^{(l)} = \text{UPDATE}^{(l)} \left( \mathbf{z}_v^{(l-1)},\ \text{AGGREGATE}^{(l)} \left( \left\{ \mathbf{z}_u^{(l-1)} : u \in \mathcal{N}_v \right\} \right) \right), \quad l = 1, \ldots L \quad (6)$$

where $\mathbf{z}_v^{(0)} = \mathbf{x}_v$, in that the node representation of 0-th layer is input node features. AGGREGATE $(\cdot)$ function aggregates the representations of the neighbor set with a particular function. UPDATE $(\cdot)$ function is designed to update a node reprsentation $\mathbf{z}_v^{(l-1)}$ with the aggregated information produced by AGGREGATE $(\cdot)$. With the message-passing, $L$-layer GNN provides node representations $\mathbf{z}_v^{(L)}$ that express an $L$-hop egograph given the node $v$ as a center node. In this paper, we use a pre-trained

Table 7: Performance on chemical reaction tasks, including forward reaction prediction and retrosynthesis. ∗ denotes the model fine-tuned with task-specific instruction data.

| Model | Exact↑ | BLEU↑ | Levenshtein↓ | RDK FTS↑ | MACCS FTS↑ | Morgan FTS↑ | Validity↑ |
|---|---|---|---|---|---|---|---|
| *Forward Reaction Prediction* | | | | | | | |
| Alpaca[†] [14] | 0.000 | 0.065 | 41.989 | 0.004 | 0.024 | 0.008 | 0.138 |
| Baize[†] [51] | 0.000 | 0.044 | 41.500 | 0.004 | 0.025 | 0.009 | 0.097 |
| ChatGLM[†] [52] | 0.000 | 0.183 | 40.008 | 0.050 | 0.100 | 0.044 | 0.108 |
| LLaMA[†] [53] | 0.000 | 0.020 | 42.002 | 0.001 | 0.002 | 0.001 | 0.039 |
| Vicuna[†] [37] | 0.000 | 0.057 | 41.690 | 0.007 | 0.016 | 0.006 | 0.059 |
| LLaMA∗ [53] | 0.012 | 0.804 | 29.947 | 0.499 | 0.649 | 0.407 | **1.000** |
| Mol-Instruction [48] | 0.045 | 0.654 | 27.262 | 0.313 | 0.509 | 0.262 | **1.000** |
| InstructMol-G [54] | 0.153 | 0.906 | 20.155 | 0.519 | 0.717 | 0.457 | **1.000** |
| InstructMol-GS [54] | 0.536 | **0.967** | 10.851 | 0.776 | 0.878 | 0.741 | **1.000** |
| **LLaMo (Ours)** | **0.584** | 0.894 | **6.162** | **0.857** | **0.918** | **0.841** | 0.938 |
| *Retrosynthesis* | | | | | | | |
| Alpaca[†] [14] | 0.000 | 0.063 | 46.915 | 0.005 | 0.023 | 0.007 | 0.160 |
| Baize[†] [51] | 0.000 | 0.095 | 44.714 | 0.025 | 0.050 | 0.023 | 0.112 |
| ChatGLM[†] [52] | 0.000 | 0.117 | 48.365 | 0.056 | 0.075 | 0.043 | 0.046 |
| LLaMA[†] [53] | 0.000 | 0.036 | 46.844 | 0.018 | 0.029 | 0.017 | 0.010 |
| Vicuna[†] [37] | 0.000 | 0.057 | 46.877 | 0.025 | 0.030 | 0.021 | 0.017 |
| LLaMA∗ [53] | 0.000 | 0.283 | 53.510 | 0.136 | 0.294 | 0.106 | **1.000** |
| Mol-Instruction [48] | 0.009 | 0.705 | 31.227 | 0.283 | 0.487 | 0.230 | **1.000** |
| InstructMol-G [54] | 0.114 | 0.586 | 21.271 | 0.422 | 0.523 | 0.285 | **1.000** |
| InstructMol-GS [54] | **0.407** | **0.941** | 13.967 | 0.753 | 0.852 | 0.714 | **1.000** |
| **LLaMo (Ours)** | 0.341 | 0.830 | **12.263** | **0.793** | **0.868** | **0.750** | 0.954 |

GIN [21] with 5 layers for the graph encoder, which has widely been applied to molecule graph understanding tasks [4]. Formally, GIN updates node representations $\mathbf{z}_v^{(l)}$ with the following equation:

$$\mathbf{z}_v^{(l)} = \text{MLP}^{(l)}\left(\left(1 + \epsilon^{(l)}\right) \cdot \mathbf{z}_v^{(l-1)} + \sum_{u \in \mathcal{N}_v} \mathbf{z}_u^{(l-1)}\right), \tag{7}$$

where $\epsilon^{(l+1)}$ is a learnable parameter or a fixed scalar. The aggregate function $\text{AGGREGATE}(\cdot)$ is the sum operation and the MLP function is used for the combine function $\text{UPDATE}(\cdot)$.

## D   Additional Experimental Results

We also conduct experiments to validate the effectiveness of our LLaMo on chemical reaction prediction, such as forward reaction prediction and retrosynthesis tasks in Table 7. The table demonstrates that our LLaMo still performs well on the chemical reaction tasks. LLaMo achieves the best performance on Levenshtein, RDK FTS, MACCS FTS, and Morgan FTS metrics across diverse baselines in all tasks, which indicates that LLaMo successfully comprehends molecular graph structure. We conjecture that our multi-level graph projector helps the large language model understand the molecular graph structure by representing multi-hop structural information to the molecular graph tokens.

## E   Detailed Experimental Settings

### E.1   Implementation Details

Our code is implemented based on PyTorch [55] library. Also, we adopt PyTorch Geometric (PyG) [56], and Huggingface transformers [57] to utilize the graph architectures and Large Language Models (LLMs). PEFT [58] and OpenDelta [59] libraries are used for parameter-efficient fine-tuning of LLMs, *i.e.*, LoRA. We use LLaMA 2 chat 7B model [9] and Galactica 1.3B [2] as our base language model. We leverage GIN [21] with five layers initialized based on the MoleculeSTM graph encoder [60], which is pre-trained with text-graph contrastive learning [61]. We use LoRA to train the large language model in stage 2. We use OGB [62], a smiles2graph function, to convert SMILES representations to 2D graphs. Our experiments are run on 4 × A6000 GPUs or 4 × V100 GPUs and 2 × A6000 GPUs for LLaMA2 and Galactica, respectively. In stage 1, the AdamW [63] optimizer is adapted with an initial learning rate of 1e-4 (minimum learning rate is 1e-5 and warmup learning rate

is 1e-6). The warmup step is 1,000 and the cosine scheduler is applied. In stage 2, the initial learning rate is set to 5e-5 (minimum learning rate is 5e-6 and warmup learning rate is 5e-7).

To evaluate the efficacy of the proposed method, we fine-tune baseline models and evaluate them for three tasks such as **1)** molecule description generation, **2)** IUPAC name prediction, and **3)** property question answering. We conducted experiments under two major settings: generalist and specialist models. In the generalist setting, one model handles all three tasks, whereas in the specialist setting, we train a model for each downstream task. For generalist experiments, we use datasets derived from PubChem and QM9. To be specific, for 'molecule description generation', and 'property prediction', we use the datasets derived from PubChem and QM9 of MoleculeNet [64] as in Mol-Instructions [48]. For IUPAC name prediction, a dataset derived from [3] is used. To train the generalist variant of LLaMo, we use a training split of molecular description generation dataset of Mol-Instructions in stage 1. In stage 2, the model is instruction-tuned with a training split of description generation and property prediction instruction dataset of Mol-Instructions, IUPAC name prediction from [3], and our GPT-generated instruction-following data. For the evaluation of molecular description generation and property question answering tasks, we use the test split of Mol-Instructions molecular description generation and property prediction datasets, which are sampled from PubChem [44] and QM9 dataset of MoleculeNet [64], respectively. For ablations, we use a short training schedule (epoch 1 pre-training, epoch 1 instruction tuning). For the final models, we adopt a long training schedule (epoch 1 pre-training, epoch 3 instruction tuning).

For training the specialist variant of LLaMo, we follow MolCA [3] to train the model with the pretrain split of PubChem324k in the stage 1 phase and fine-tune the model for each specific downstream task in stage 2. For the inference under specialist experiments, where a model is individually finetuned for a specific downstream task, we use a test split of PubChem324k [3], ChEBI-20 [29] and IUPAC name prediction dataset from [3].

## E.2 Metrics

We report BLEU [65] and METEOR [66] for the molecule description generation and IUPAC name prediction tasks. MAE is reported for property QA.

**BLEU.** The BLEU metric measures the quality of generated text by comparing $n$-gram sequence between the generated text and the reference text, which can be formulated as:

$$\text{BLEU} = \text{BP} \times \exp\left(\frac{1}{N}\sum_{n=1}^{N}\log p_n\right), \tag{8}$$

where $N$ is the number of $n$-grams and $p_n$ is the precision, *i.e.*, the ratio of the number of $n$-grams in the generated text appearing in the reference text. The BLEU score also takes into account sequence length with Brevity Penalty (BP) as:

$$\text{BP} = \begin{cases} 1 & \text{if } c > r \\ e^{(1-r/c)} & \text{if } c \leq r \end{cases}, \tag{9}$$

where $c$ and $r$ are the lengths of generated and reference texts, respectively. This encourages the model to avoid generating short sequences. In our experiments, we use BLEU-4 as the default BLEU metric.

**METEOR.** The METEOR metric is proposed to consider both precision and recall between the generated text and the reference text, which is as follows:

$$P = \frac{\text{number of matched words}}{\text{number of words in generated text}}, \quad R = \frac{\text{number of matched words}}{\text{number of words in reference text}}, \tag{10}$$

$$F = \frac{10PR}{9P + R}, \tag{11}$$

$$\text{Penalty} = 0.5 \cdot \left(\frac{\text{number of chunks}}{\text{number of matched words}}\right), \tag{12}$$

$$\text{METEOR} = F \cdot (1 - \text{Penalty}), \tag{13}$$

where a chunk is a set of uni-grams which are adjacent in the generated text and in the reference text. Similar to BLEU, the Penalty is reflected to take into account the length of generated text. Therefore, the METEOR metric is specialized to measure the morphology, fluency, and adequacy of text rather than sequence order since it does not use $n$-grams.

**MAE.** Mean Absolute Error (MAE) aims to measure the average magnitude of errors between the ground-truth values and predicted values, which is defined as:

$$\text{MAE} = \frac{1}{N} \sum_{n=1}^{N} |\hat{y}_n - y_n|. \tag{14}$$

where $y_n$ is a ground-truth and $\hat{y}_n$ is a model prediction.

# F   Baselines

For the generalist models, we train our LLaMo based on `Llama-2-7b-chat` [9] as a backbone language model for a fair comparison with Mol-Instructions [48]. We compare our LLaMo with (1) LLM-based generalist models including Galactica [2], `Llama2-7b-chat` [9], GPT-3.5, and GPT-4, (2) Molecule instruction-tuned generalist model such as Mol-Instructions [48]. Since GPT-3.5 and GPT-4 have difficulty in solving the tasks without in-context learning, we additionally measure the performance of GPT-3.5 and GPT-4 with 4-shots in-context learning, which are GPT-3.5 (ICL) and GPT-4 (ICL). The 4-shot exemplars are selected by computing the Tanimoto similarity [67] using a 2048-bit Morgan Fingerprint [68] with RDKit[2], choosing the four molecules most similar to the target molecule from the train split of each dataset. For the specialist models, we train our LLaMo with Galactica 1.3B [2] for a fair comparison with MolCA [3]. We use single-task specialist models as baselines, including MolT5 [29], MoMu [4], and MolCA [3].

# G   Benchmarks

In this section, we provide a brief introduction to each task and dataset utilized in our research.

**Molecular description generation.** Generating the description of a molecule is considered one of the most important tasks in molecular language models. For a given molecule, we aim for the model to generate an accurate and informative description including various chemical properties, functional groups, biological roles, and real-world applications of the molecule. Developing a model capable of generating such descriptions is highly valuable because it has the potential to discover information about molecules that is currently unknown or very expensive to find out, thus serving as a powerful assistant for various tasks in chemistry and biology. Therefore, various works [3, 4, 29, 69, 70, 71] have tried to enhance the ability to generate appropriate descriptions of chemical compounds using language models or multi-modal language models.

For testing the performance of molecule description generation of LLaMo and previous models, we utilize the molecular description generation dataset of Mol-Instructions[3] [48], based on PubChem database [44] for generalist models, and both PubChem324k [3] and ChEBI-20 [8] for specialist models. PubChem324k is constructed by collecting 324k molecules and their associated text information from the PubChem database. ChEBI-20 is the most commonly utilized benchmark in this task, consisting of selected 33,010 pairs of molecules and descriptions from ChEBI [72]. Each description of ChEBI-20 contains more than 20 words and includes various and rich information about molecules, such as conjugate base/acid, functional parent, and enantiomer of molecules. We employ a test split of each dataset: 1,000 samples of Mol-Instructions and 2,000 / 3,300 samples of PubChem324k / ChEBI-20. Similarly, we filter out samples of which SMILES representation cannot be converted into the molecular graph via RDKit.

**IUPAC name prediction.** IUPAC (International Union of Pure and Applied Chemistry) nomenclature [73] provides a systematic naming convention for molecules based on pre-defined rules, eliminating the ambiguity in molecular names. It is the standard method for naming molecular

---

[2]`https://github.com/rdkit/rdkit`. Copyright (c) 2006-2015, Rational Discovery LLC, Greg Landrum, and Julie Penzotti and others. Licensed under BSD 3-Clause license

[3]Copyright (c) 2023 ZJUNLP. Licensed under CC-BY 4.0 license

compounds in chemistry, and as such, numerous pieces of chemical literature utilize IUPAC names to represent molecules. Consequently, previous studies [2, 3] have adopted the task of predicting IUPAC names from SMILES representations to evaluate the chemical understanding capability of molecular language models. We also follow this approach in our research. To assess the performance of LLaMo and baseline models, we use the PubChem324k dataset [3] again. The test split of PubChem324k offers curated high-quality samples of 2,000 molecules, each with corresponding SMILES representations and IUPAC names. We provide the SMILES representation of each molecule to the model along with a prompt and then compare the model-generated IUPAC names with the ground-truth IUPAC names to evaluate performance.

**Property prediction.** In chemical and biological domains, particularly in drug discovery, exploring chemical compounds that satisfy specific chemical properties is crucial. Therefore, the ability to estimate the chemical and physical properties is essential for chemical foundation models. Additionally, leveraging machine learning-based approaches for predicting molecular properties has proven to be much more efficient than traditional approaches in computational chemistry. Inspired by these facts, property prediction with LLMs has recently drawn attention from researchers [2, 69, 70, 71]. So, we also assess the property QA performance of LLaMo and baselines using the dataset introduced by [48]. This dataset is a subset of QM9 dataset from MoleculeNet [64], a widely used benchmark for various chemistry machine learning tasks. The original QM9 dataset contains numerical values for 19 chemical properties per molecule, but [48] focuses on three properties related to molecular orbital energy: HOMO (Highest Occupied Molecular Orbital) energy, LUMO (Lowest Unoccupied Molecular Orbital) energy, and the HOMO-LUMO gap (in Hartree units). [48] also created distinct question-form instructions for each property to make language models understand the task and accurately generate continuous values. Consequently, using the given instruction for each molecule in the 2,000 test samples and its SMILES representation as input, we have LLaMo and baseline generalist models predict the properties of the given molecules. Similarly, we filter out samples of which SMILES representation cannot be converted into the molecular graph via RDKit.

**Forward reaction prediction.** Understanding chemical reactions is crucial for advancing various chemical and biological fields, including pharmaceuticals, materials science, and environmental technology. This understanding enables researchers to design efficient synthesis pathways and develop new chemical compounds. Therefore, if a molecular model can effectively comprehend these chemical reactions, it can serve as a powerful assistant in the research and development process. This concept has emerged through the task of forward reaction prediction, and several prior models [48, 54] have conducted experiments to address this task. This task focuses on the proactive determination of potential products resulting from a chemical reaction based on given reactants and reagents. This approach is also significant from the perspectives of efficiency and environmental sustainability, as it reduces the need for experimental trial and error in chemical development and research by using these models to virtually conduct simulated experiments. We utilize the forward reaction prediction segment of Mol-Instructions [48], based on the USPTO [74] database, to assess the performance of LLaMo and other models. This dataset contains question-form instructions for predicting reaction products, with reactants and reagents separated by a period ('.') as input, and the corresponding product of the reaction as the target output. It also doesn't specify what the reactants and reagents are to create a task that more closely resembles real-world scenarios. Finally, using the test samples from this dataset, we provide the SMILES representations of the reactants and reagents to the model and their molecular graphs with a prompt, aiming to predict the product molecule in SMILES format as well.

**Retrosynthesis.** Retrosynthesis shares a similar context with forward reaction prediction but approaches chemical reactions from a different perspective. The retrosynthetic analysis begins with the target compound and works backward to identify potential reactant molecules for its synthesis. This reverse approach is as valuable as forward reaction synthesis because it aids researchers in discovering effective and efficient synthetic methodologies for generating target molecules. This is particularly important in various chemical applications, such as drug discovery, where identifying chemically valid and economical processes for synthesizing target drug molecules is essential. As a result, several previous studies [48, 54] have adopted this task to analyze the capability to deeply understand chemical knowledge and take advantage of this knowledge to specify the chemical reactants of a given product. We also follow this approach and demonstrate its capability using the retrosynthesis section of the Mol-Instructions [48] dataset, which originates from the USPTO_500MT [75]. Each sample consists of a product molecule and the required reactant molecules, separated by a period ('.').

Table 8: Example prompt for GPT in molecular description generation task. The top block shows the instruction part of the prompts. It includes original instruction from the dataset and additional instruction added only for in-context learning. The bottom block shows optional few-shot examples and target SMILES.

> You are an expert chemist. Please strictly follow the format, no other information can be provided. Given the molecular SMILES, could you provide a description of this molecule? You will be provided with several examples of molecules and their descriptions.
> ________________________________________________
> Molecule SMILES: COc1ccc(-c2cc(=O)c3c(O)c(Oc4ccc(-c5cc(=O)c6c(O)cc(O)cc6o5)cc4)c(OC)cc3o2)cc1
> Molecular Description: The molecule is a natural product found in Selaginella tamariscina, Taxodium distichum, and other organisms with data available.
>
> ...
>
> Molecule SMILES: CC=Cc1ccc2oc(-c3cccc(Oc4cc(-c5oc6ccc(C=CC)cc6c5C)ccc4O)c3)c(C)c2c1
> Molecular Description: The molecule is a natural product found in Piper aequale with data available.
>
> Molecule SMILES: C/C=C/c1ccc2oc(-c3ccc(Oc4cc(-c5oc6ccc(/C=C/C)cc6c5C)ccc4O)c3)c(C)c2c1
> Molecular Description:

Table 9: Example prompt for GPT in IUPAC name prediction task. The top block shows the instruction part of the prompts. It includes additional instruction added only for in-context learning. The bottom block shows optional few-shot examples and target SMILES.

> You are an expert chemist. Please strictly follow the format, no other information can be provided. Given the molecular SMILES, your task is to predict the IUPAC name using your experienced chemical IUPAC name knowledge. You will be provided with several examples of molecules and their IUPAC names.
> ________________________________________________
> Molecule SMILES: COc1cc([C@H]2COc3cc(O)ccc3C2)ccc1O
> The molecule's IUPAC name is (3S)-3-(4-hydroxy-3-methoxyphenyl)-3,4-dihydro-2H-chromen-7-ol
>
> ...
>
> Molecule SMILES: COc1c([C@@H]2COc3cc(O)ccc3C2)ccc2c1C=CC(C)(C)O2
> The molecule's IUPAC name is (3R)-3-(5-methoxy-2,2-dimethylchromen-6-yl)-3,4-dihydro-2H-chromen-7-ol
>
> Molecule SMILES: COC1=CC(=O)C(C2COc3cc(O)ccc3C2)=CC1=O
> The molecule's IUPAC name is

For each test sample, we provide the product's SMILES representation, its molecular graph, and the given instruction prompt as input to the model to generate possible reactants in SMILES format.

# H Prompts

In this section, we provide input prompts used to evaluate GPT-3.5 and GPT-4 on molecular description generation task (Table 8), IUPAC name prediction task (Table 9), and property question answering task (Table 10). We also provide the input prompts for machine-generated instruction-tuning data generation (Table 11, 12).

# I Over-smoothing Problems

We provide additional samples to show the over-smoothing problems in Figure 7.

Table 10: Example prompt for GPT in property question answering task. The top block shows the instruction part of the prompts. It includes original instruction from the dataset and additional instruction added only for in-context learning. The bottom block shows optional few-shot examples and target SMILES.

> You are an expert chemist. Please strictly follow the format, no other information can be provided. Given the molecular SMILES, what is the energy separation between the HOMO and LUMO of this molecule? You will be provided with several examples of molecules and their HOMO-LUMO gap values.
> ___
> Molecule SMILES: COCC12OC3CC1C32
> Output Value: 0.2967
>
> ...
>
> Molecule SMILES: OCCC12CC3C(O1)C32
> Output Value: 0.305
>
> Molecule SMILES: CCC1C2OC3C1C23C
> Output Value:

Table 11: Input prompt for generating multi-turn conversation data based on SMILES and caption.

> You are an AI chemical assistant, and you are seeing a single molecule. What you see is provided with SMILES representation of the molecule and sentences describing the same molecule you are analyzing. Answer all questions as you are seeing the molecule.
> Ask diverse questions and give corresponding answers.
> Include questions asking about the detailed information of the molecule, including the class, conjugate acid/base, functional groups, chemical role, etc.
> Do not ask any question that cannot be answered confidently.
> Molecule SMILES: {SMILES}
> Caption: {CAPTION}
> Conversation:

Table 12: Input prompt for generating multi-turn conversation data based on SMILES, caption, and IUPAC.

> You are an AI chemical assistant, and you are seeing a single molecule. What you see is provided with SMILES representation of the molecule and sentences describing the same molecule you are analyzing. In addition, the IUPAC name of the molecule is given. Answer all questions as you are seeing the molecule.
> Ask diverse questions and give corresponding answers.
> Include questions asking about the detailed information of the molecule, including the class, conjugate acid/base, functional groups, chemical role, etc.
> Do not ask any questions that cannot be answered confidently.
> Molecule SMILES: {SMILES}
> Caption: {CAPTION}
> IUPAC: {IUPAC}
> Conversation:

## J  More Qualitative Samples

We provide further qualitative results of LLaMo w/ and w/o graph in Figure 8, and LLaMo w/ and wo MGProj in Figure 9, respectively.

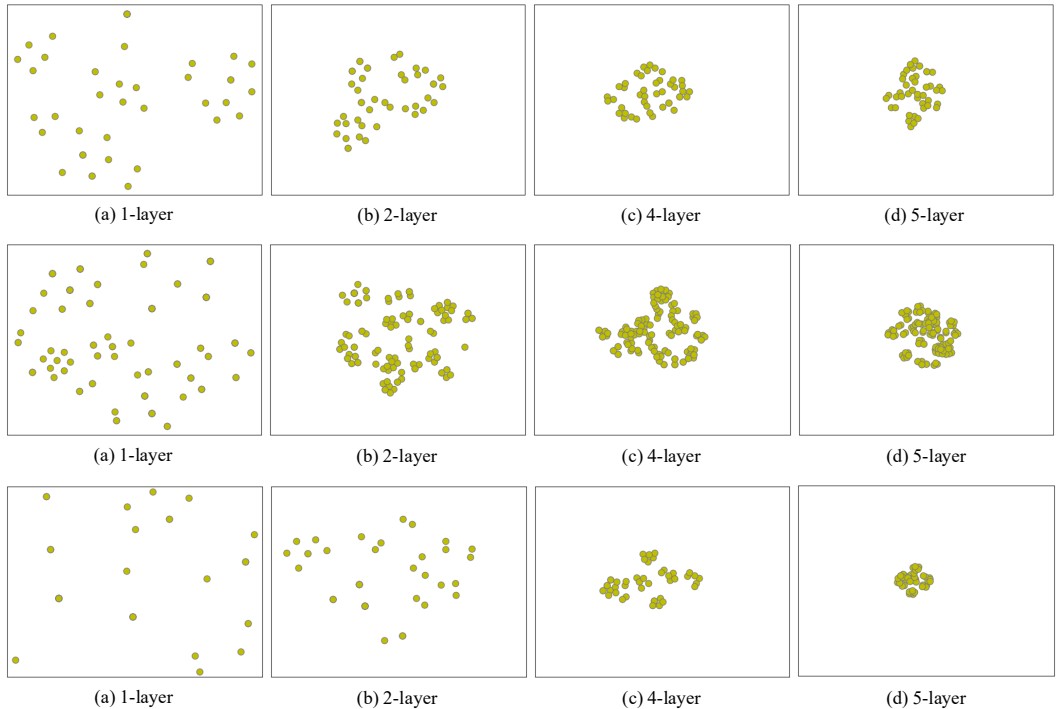

Figure 7: Node representations of graph encoder with 1,2,4,5 layers. As the number of layers increases, node representations collapse.

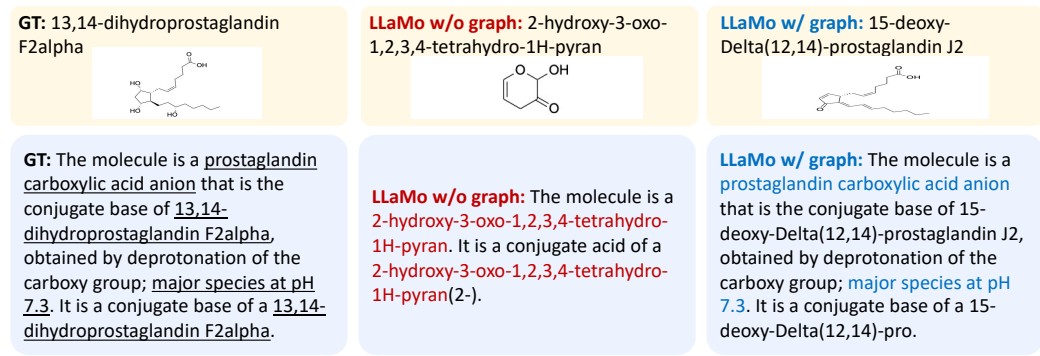

**GT:** 13,14-dihydroprostaglandin F2alpha

**LLaMo w/o graph:** 2-hydroxy-3-oxo-1,2,3,4-tetrahydro-1H-pyran

**LLaMo w/ graph:** 15-deoxy-Delta(12,14)-prostaglandin J2

**GT:** The molecule is a prostaglandin carboxylic acid anion that is the conjugate base of 13,14-dihydroprostaglandin F2alpha, obtained by deprotonation of the carboxy group; major species at pH 7.3. It is a conjugate base of a 13,14-dihydroprostaglandin F2alpha.

**LLaMo w/o graph:** The molecule is a 2-hydroxy-3-oxo-1,2,3,4-tetrahydro-1H-pyran. It is a conjugate acid of a 2-hydroxy-3-oxo-1,2,3,4-tetrahydro-1H-pyran(2-).

**LLaMo w/ graph:** The molecule is a prostaglandin carboxylic acid anion that is the conjugate base of 15-deoxy-Delta(12,14)-prostaglandin J2, obtained by deprotonation of the carboxy group; major species at pH 7.3. It is a conjugate base of a 15-deoxy-Delta(12,14)-pro.

Figure 8: An example of molecular description generation results of LLaMo w/o graph and LLaMo w/ graph given the molecule "CCCCC[C@@H1](CC[C@H1]1[C@@H1](C[C@@H1]([C@@H1] 1C/C=C\CCCC(=O)[O-1])O)O)O)".

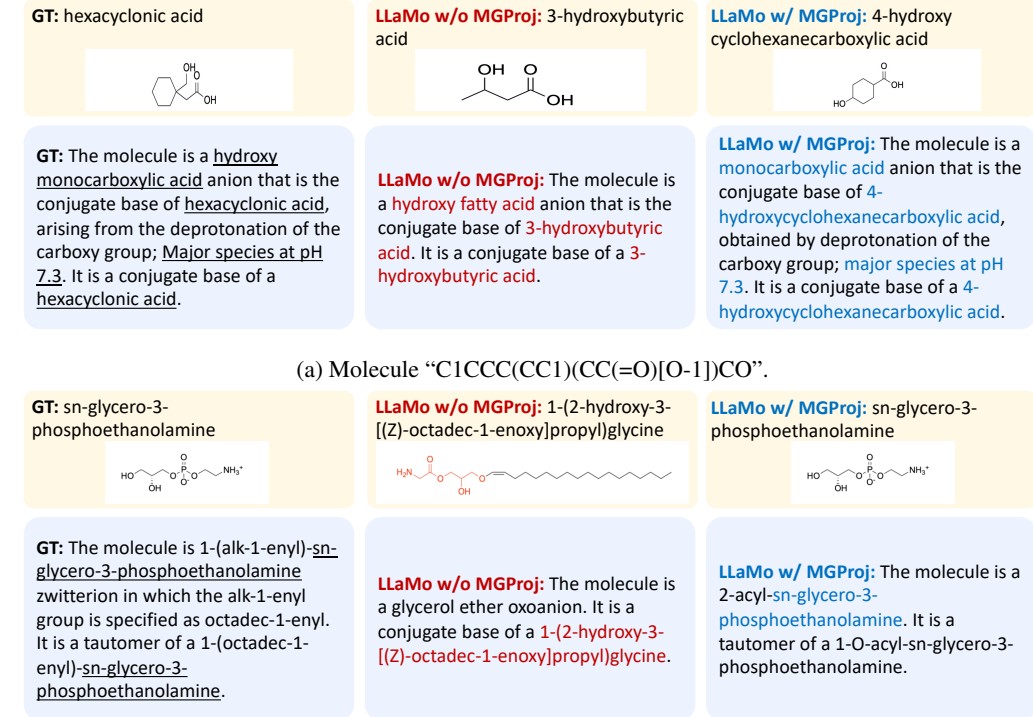

**GT:** hexacyclonic acid

**LLaMo w/o MGProj:** 3-hydroxybutyric acid

**LLaMo w/ MGProj:** 4-hydroxy cyclohexanecarboxylic acid

**GT:** The molecule is a hydroxy monocarboxylic acid anion that is the conjugate base of hexacyclonic acid, arising from the deprotonation of the carboxy group; Major species at pH 7.3. It is a conjugate base of a hexacyclonic acid.

**LLaMo w/o MGProj:** The molecule is a hydroxy fatty acid anion that is the conjugate base of 3-hydroxybutyric acid. It is a conjugate base of a 3-hydroxybutyric acid.

**LLaMo w/ MGProj:** The molecule is a monocarboxylic acid anion that is the conjugate base of 4-hydroxycyclohexanecarboxylic acid, obtained by deprotonation of the carboxy group; major species at pH 7.3. It is a conjugate base of a 4-hydroxycyclohexanecarboxylic acid.

(a) Molecule "C1CCC(CC1)(CC(=O)[O-1])CO".

**GT:** sn-glycero-3-phosphoethanolamine

**LLaMo w/o MGProj:** 1-(2-hydroxy-3-[(Z)-octadec-1-enoxy]propyl)glycine

**LLaMo w/ MGProj:** sn-glycero-3-phosphoethanolamine

**GT:** The molecule is 1-(alk-1-enyl)-sn-glycero-3-phosphoethanolamine zwitterion in which the alk-1-enyl group is specified as octadec-1-enyl. It is a tautomer of a 1-(octadec-1-enyl)-sn-glycero-3-phosphoethanolamine.

**LLaMo w/o MGProj:** The molecule is a glycerol ether oxoanion. It is a conjugate base of a 1-(2-hydroxy-3-[(Z)-octadec-1-enoxy]propyl)glycine.

**LLaMo w/ MGProj:** The molecule is a 2-acyl-sn-glycero-3-phosphoethanolamine. It is a tautomer of a 1-O-acyl-sn-glycero-3-phosphoethanolamine.

(b) Molecule "CCCCCCCCCCCCCCCCCC=COC[C@H1](COP(=O)([O-1])OCC[NH3+1])O".

Figure 9: Examples of molecular description generation results of LLaMo w/o MGProj and LLaMo w/ MGProj.

