# OpenReview forum: "LLaMo: Large Language Model-based Molecular Graph Assistant"
_NeurIPS.cc/2024/Conference — NeurIPS 2024 poster_

### Official Review · Reviewer_3eLt · 2024-06-26

**Soundness:** 3
**Presentation:** 3
**Contribution:** 2
**Rating:** 6
**Confidence:** 3

**Summary:**

The paper introduces LLaMo, a Large Language Model-based Molecular graph assistant. LLaMo is a model that integrates a GNN encoder, a multi-level graph projector, and a language model. The projector uses a cross-attention mechanism to convert graph representations into graph tokens by abstracting outputs from each GNN layer and motif representations. Additionally, the authors use machine-generated molecular graph instruction data, leveraging GPT-4, for instruction-tuning. Extensive experiments show that LLaMo excels in some tasks.

**Strengths:**

The framework incorporates a multi-level graph projector that effectively captures multi-scale information. This design mitigates the over-smoothing problem. A figure is provided to illustrate the significant impact of this design on model performance and attention distribution.

The paper offers a comprehensive, step-by-step explanation of each component within the LLaMo framework. Additionally, a lot of typical examples are presented to clearly demonstrate the framework's design and functionality in practical applications.

**Weaknesses:**

In order to perform instruction-tuning, the authors utilized GPT-4 to generate molecular graph- text instruction-following data using graph-text pair datasets with human-written instructions. Although this approach enhances the results, it lacks a robust evaluation of the generated data's quality and validity. It is crucial to develop methodologies to rigorously assess the accuracy and relevance of the GPT-4 generated instruction data to ensure the reliability of the model's performance improvements.

**Questions:**

The two-step training process involves initially training the graph encoder in Stage 1, followed by fine-tuning the Large Language Model using LoRA in Stage 2. Why is the multi-level projector set as trainable during both training stages?

**Limitations:**

The experiments predominantly concentrate on a few tasks such as molecular description generation, property prediction, and IUPAC name prediction. This limited focus may not fully reveal the model’s versatility and potential shortcomings across a wider spectrum of molecular tasks.

---

> ### Author Rebuttal · Authors · 2024-08-07
>
> **[W1] Evaluation of GPT-generated quality.**
>
> Thank you for your constructive feedback. During the generation, we implement a multi-step assessment process, detailed below:
>
> Step 1: We prompt GPT-4 to generate multi-turn conversation instruction data about molecules using captions and IUPAC names from a well-established dataset, PubChem, without any demonstrations (zero-shot).
>
> Assessment 1: We sampled over 100 subsets of data and observed that GPT-4 frequently generated **incomplete conversations and refused generation.**
>
> Step 2: To address these issues, we first sample high-quality demonstrations from a small set of complete conversations generated by GPT-4 (zero-shot). Subsequently, we prompt GPT-4 to generate data with these demonstrations.
>
> Assessment 2: We sampled 500 subsets generated via in-context learning. We found that conversations with a higher number of turns were more **prone to generating incomplete and inaccurate outputs**.
>
> Step 3: We filter out incomplete conversations and those with many turns. Approximately 5% of the data was filtered out.
>
> Assessment 3: We sampled 500 subsets from the filtered data and manually assessed their quality. We verified that the generated data contained accurate information about the given molecule, with no issues of incompleteness.
>
> In addition, our ablation studies (Table 5) validate the effectiveness of GPT-4 generated instruction dataset. Instruction tuning with this data improves the performance of LLaMo, providing the model with more detailed and instruction-following guidance.
>
> We believe that our data is high-quality and will make our instruction data publicly available for future research in the molecular domain.
>
> **[Q1] Why is the multi-level projector set as trainable during both stages?**
>
> We set the projector as trainable both stage 1 and stage 2 to follow existing Large Vision-Language Models such as LLaVA [1].
>
> **[Limitations1] Other tasks.**
>
> Thank you for the suggestion. We conduct additional experiments on the forward reaction prediction task to evaluate LLaMo’s ability in other spectrum of molecular tasks, as suggested. The forward reaction prediction is the task that targets the prediction of chemical reactions given the reactants. We utilize the forward reaction prediction dataset from Mol-Instructions. The experimental results are in the table below. From the table, our LLaMo shows the best performance compared to other models in all metrics.
>
> | Model | BLEU&uarr;| Levenshtein dist.&darr;| Tanimoto Sim.&uarr;|
> | --- | --- | --- | --- |
> | Galactica | 46.8 | 35.02 | 0.16 |
> | Text+Chem T5 | 78.2 | 20.41 | 0.71 |
> | Mol-Instructions | 65.4 | 27.26 | 0.31 |
> | LLaMo | 82.4 | 6.81 | 0.76 |
>
> [1] Liu, Haotian, et al. "Visual instruction tuning." NeurIPS 2023.

---

> > ### Comment · Reviewer_3eLt · 2024-08-09
> >
> > I thank the authors for the rebuttal. It addressed most of my concerns and I tend to keep my score.

---

> ### Author Response · Authors · 2024-08-11
>
> We sincerely appreciate the reviewer's comments and positive score.
>
> As the reviewer suggested, we will include the discussion on the quality of GPT-generated data in the final version.

---

### Official Review · Reviewer_W78D · 2024-07-05

**Soundness:** 3
**Presentation:** 4
**Contribution:** 3
**Rating:** 7
**Confidence:** 4

**Summary:**

This paper proposes a molecule-text model, LLaMo, which aligns text and molecule modalities to tackle diverse downstream tasks with a language interface. Through two-stage alignment: training a graph encoder and then tuning the LLM through instructions, LLaMo aligns text and molecular representations well, and achieves promising performance on diverse downstream tasks.

**Strengths:**

1. The writing and presentation of this paper are clear. It is well-motivated and easy to read.
2. Code implementation and comprehensive details are provided, which ensures reproducibility.
3. The design of learnable query tokens is interesting.
4. I observe an interesting phenomenon in the experiments. Low-level information is sometimes overlooked for graph-level prediction, and it is mainly studied for node-level predictions. However, for molecule-text models, some tasks require both local and global-level information, and it presents more importance for tasks like IUPAC prediction.

**Weaknesses:**

1. The implementation details of functional groups are not given in the main text nor appendix. Could you share more details on this part?
2. How is the graph modality treated in the instruction tuning phase? Is it fixed, and the graph token is augmented into the text token inputs?
3. I am curious about the design of the learnable query tokens. Why are the motif and graph-level representations modeled separately?
4. One factor that prevents me from giving a higher score is that this pipeline is widely adopted for vision-language models, so it does not surprise me.

**Questions:**

Please refer to the questions given in the weakness part

**Limitations:**

Limitations and broader impacts are discussed in the appendix. Guidelines should be removed from the checklists.

---

> ### Author Rebuttal · Authors · 2024-08-07
>
> **[W1] Details of functional group representations.**
>
> We appreciate your feedback and provide a comprehensive explanation regarding functional representations. We use simple hand-crafted features derived from the molecular graph as functional group information.
>
> To construct functional group representations, we initially identify functional groups from the given molecular graph following [1]. We define three types of substructures (rings, non-cyclic FGs, and carbon-carbon single bonds) as functional groups.
>
> Then, we vectorize the main characteristics of each functional group. Specifically, we one-hot encode the number of important atoms (e.g., carbon, oxygen, etc) and bonds (single, double, triple, aromatic bonds) contained in each functional group to represent the functional group. Each functional group is represented as $\mathbf{z}_{\text{FG},i}$, a vector encoding of its main characteristics.
>
> Finally, the functional group representations $\mathbf{Z}\_{\text{FG}}$ are constructed by concatenating all individual functional group representations $\mathbf{z}\_{\text{FG}, i}$, which is formulated as $\mathbf{Z}\_{\text{FG}} = \left[\mathbf{z}\_{\text{FG},0}, \dots, \mathbf{z}\_{\text{FG},M}\right],$ where $M$ indicates the number of the functional groups in the given molecular graph.
>
> We hope that this explanation clarifies the details of the functional groups. We will include these details in our camera-ready version if the paper gets accepted.
>
> **[W2] How is the graph modality treated in the instruction tuning phase?**
>
> As mentioned in Section 3.2, in the instruction tuning phase, the input molecular graph is first represented with a sequence of graph tokens and then the sequence of graph tokens is augmented into the text token inputs. In this phase, we train both multi-level graph projector and LLM, while keeping the graph encoder frozen.
>
> **[W3] Why are the motif and graph-level representations modeled separately?**
>
> Graph-level representations $\hat{\mathbf{P}}^{(0)}, \dots \hat{\mathbf{P}}^{(L)}$ are constructed **based on outputs of each layer of GNNs**, whereas motif-level representations $\hat{\mathbf{P}}^{(\text{motif})}$ are constructed based on **functional group representations** $\mathbf{Z}_{\text{FG}}$ as detailed in **[W1] Details of functional groups.** Due to these distinct construction methods, we model them separately.
>
> **[W4] Comparisons with vision-language model pipelines.**
>
> Compared to the Large Vision-Language Models (LVLMs) such as LLaVA [2], Instruct-BLIP [3], and mPLUG-owl [4], etc., our proposed LLaMo has several unique components tailored to the molecule domain, as summarized below:
>
> - **Multi-level graph projector**: While most LVLMs usually construct visual tokens **solely based on the outputs from the final layer** of a visual encoder, our multi-level graph projector leverages **node representations from all layers of a GNN** to construct molecular graph tokens. This method enables the tokens to encapsulate richer information, reflecting the molecular graph structure at multiple levels of abstraction. This multi-level approach has been shown to improve the performance in various molecular graph-related tasks.
> - **Molecular graph specialized instruction data**: We also present **molecular graph specialized multi-turn instruction data** automatically generated by GPT-4. This specialized multi-turn instruction data aims to improve the performance of the model in tasks related to molecular graphs by providing detailed and more contextually relevant examples. We believe that this data contributes to future research concerning the molecule domain.
> - **Functional groups (Motifs)**: LLaMo **integrates molecule-specialized functional group information**. The functional groups are statistically important subgraph patterns within a molecule exhibiting consistent chemical behaviors across different compounds. By incorporating the functional groups, our LLaMo has shown the best performance in Table 2 and 3. We have also demonstrated the effectiveness of the functional groups in Table 4.
>
> We hope that this summarization clarifies the uniqueness of our work compared to the existing LVLM works.
>
> [1] Ji, Zewei, et al. "ReLMole: Molecular representation learning based on two-level graph similarities." *Journal of Chemical Information and Modeling* 2022.
>
> [2] Liu, Haotian, et al. "Visual instruction tuning." NeurIPS 2023.
>
> [3] Dai, Wenliang, et al. "Instructblip: Towards general-purpose vision-language models with instruction tuning." NeurIPS 2023.
>
> [4] Ye, Qinghao, et al. "mplug-owl: Modularization empowers large language models with multimodality." arXiv 2023.

---

> > ### Comment · Reviewer_W78D · 2024-08-07
> >
> > Thanks for the clarification and additional results. I have raised my score to $7$.

---

> > > ### Author Response · Authors · 2024-08-11
> > >
> > > We sincerely appreciate the reviewer's positive comments and increased rating.
> > >
> > > As the reviewer suggested, we will provide more details of the functional group representation in the final version.

---

### Official Review · Reviewer_rvcw · 2024-07-09

**Soundness:** 2
**Presentation:** 3
**Contribution:** 2
**Rating:** 7
**Confidence:** 3

**Summary:**

This paper proposes a Large Language Model-based Molecular graph assistant (LLaMo), which can enhance the molecular graphs's general-purpose understanding and generation capabilities. By integrating molecular graph encoders with large language models, LLaMo enables instruction-following responses in the molecular domain. The paper conducts extensive empirical studies and justifies the effectiveness of the proposed method.

**Strengths:**

- The proposed LLaMo has a clear and effective design, making it powerful in molecular description and property prediction.
- Extensive experiments have been conducted to provide a good insight into the components of the proposed method.
- The paper is generally well-written, with clear illustrations and tables.

**Weaknesses:**

- The multi-level graph projector shares an idea similar to JKNet[1]. In addition, Graph Transformers [2] can also handle the over-smoothing problem, especially for small molecules. The author could consider discussing how the proposed techniques differ from these referenced works.
- The paper uses function groups as prior information for training; would this cause information leakage? The downstream task is to predict these groups or generate molecular captions based on these motifs.
- The GPT-4 generates samples for instruction-tunning, but the samples's qualities are not assessed properly.

[1] Representation Learning on Graphs with Jumping Knowledge Networks. In ICML, 2018.

[2] Representation Learning on Graphs with Jumping Knowledge Networks. In NeurIPS, 2022.

**Questions:**

1. Could GPT -4 perform better with prior information instilled, such as function groups?
2. What is the purpose of using different LLM for LLaMo tasks (Tables 2 and 3)? Could better LLM result in better performance? Would there be some upper bounds?
3. Could LLaMo design or edit SMILES based on the user's instructions?

**Limitations:**

The idea is novel and effective. It would be better if the author could make LLaMo able to design or edit SMILES based on the user's instructions.

---

> ### Author Rebuttal · Authors · 2024-08-07
>
> **[W1] Discussion of how multi-level graph projector differs from JKNet and Graph Transformers.**
>
> We appreciate your constructive comment. Our multi-level graph projector differs from JKNet and Graph Transformers (GTs) in two aspects: main purpose and architecture.
>
> *Main purpose*
>
> As shown in Figure 1, our multi-level graph projector functions as a **projector (yellow box)** that converts the graph encoder’s output representations into graph tokens. In contrast, both JKNet and GTs are designed as **graph encoders (green box)**. Thus, our multi-level graph projector is orthogonal to these encoders and can be used in conjunction with them.
>
> *Architecture*
>
> - JKNet
>
> Our projector encodes each graph level into **separate tokens**, while JKNet aggregates all GNN layers into **a single representation**. The separated level-aware tokens allow the attention module in the LLM to softly select the appropriate graph level, leading to better **performance (Table 4, Multi-level graph projector (MGProj) v.s. JKNet with MLP projector (MLP w/ concat)) and interpretability (Figure 4)**.
>
> - GTs
>
> Our projector does not modify the GNN architecture, unlike GTs, which adopt **self-attention operations** instead of GNNs.
>
> We will include this discussion in the camera-ready version if the paper gets accepted.
>
> **[W2] Would the use of functional groups as prior information cause information leakage?**
>
> We use **simple hand-crafted features derived from the molecular graph** as functional group information to minimize the potential risk of information leakage. They are **not tailored to specific datasets or tasks**. Specifically, we represent the functional group through one-hot vector encoding the number of key atoms (e.g., carbon) and bonds (e.g., single bond) in each substructure (e.g., rings). This representation relies solely on the inherent structure of the molecular graph.
>
> Also, we check the performance improvement by the functional groups in our ablation study of the functional group (Table 4). The table indicates that our LLaMo without motif (functional group) still shows good performance (-0.6 BLEU, +0.2 METEOR). This suggests that the performance improvements are primarily attributed to the architecture and learning scheme of LLaMo, rather than the use of functional group information.
>
> **[W3] Quality of GPT-4 generated samples.**
>
> Thank you for your constructive feedback. During the generation, we implement a multi-step assessment process, detailed below:
>
> Step 1: We prompt GPT-4 to generate multi-turn conversation instruction data about molecules using captions and IUPAC names from a well-established dataset, PubChem, without any demonstrations (zero-shot).
>
> Assessment 1: We sampled over 100 subsets of data and observed that GPT-4 frequently generated **incomplete conversations and refused generation.**
>
> Step 2: To address these issues, we first sample high-quality demonstrations from a small set of complete conversations generated by GPT-4 (zero-shot). Subsequently, we prompt GPT-4 to generate data with these demonstrations.
>
> Assessment 2: We sampled 500 subsets generated via in-context learning. We found that conversations with a higher number of turns were more **prone to generating incomplete and inaccurate outputs**.
>
> Step 3: We filter out incomplete conversations and those with many turns. Approximately 5% of the data was filtered out.
>
> Assessment 3: We sampled 500 subsets from the filtered data and manually assessed their quality. We verified that the generated data contained accurate information about the given molecule, with no issues of incompleteness.
>
> In addition, our ablation studies (Table 5) validate the effectiveness of GPT-4 generated instruction dataset. Instruction tuning with this data improves the performance of LLaMo, providing the model with more detailed and instruction-following guidance.
>
> We believe that our data is high-quality and will make our instruction data publicly available.
>
> **[Q1] Performance of GPT-4 with prior information.**
>
> Good question. We test GPT-4's performance by adding the molecule's functional group information (FG info.) to the input prompts. We use the same FG info. used in our model. The results (below table) show that adding FG info. does not consistently improve GPT-4’s performance, which means that it is not always helpful on the molecular description task.
>
> |Model|BLEU|METEOR|
> |---|---|---|
> |GPT-4|0.8|16.7|
> |GPT-4 (ICL)|27.0|52.2|
> |GPT-4 + FG info.|0.5|16.8|
> |GPT-4 (ICL) + FG info.|24.8|50.0|
> |LLaMo|37.8|63.2|
>
> **[Q2] Purpose of using different LLM on Table 2 and Table 3. Could better LLM result in better performance?**
>
> Table 2 and Table 3 report the performance of generalist and specialist models, respectively. The best-performing baseline for the generalist model, Mol-Instruction, uses LLaMa2 as its base model, while the best baseline for the specialist model, MolCA uses Galactica-1.3B. Thus, for a fair comparison with the established baselines, we employ different LLMs in each table.
>
> To study whether a better LLM improves performance, we conduct additional experiments with LLaMo using LLaMa and LLaMa2 under the same setting as in Table 2. The result (below table) shows that LLaMa2 achieves a 6.9-point improvement on the BLEU metric compared to LLaMa, meaning that a better LLM results in better performance.
>
> |Base LLM|BLEU|METEOR|
> |---|---|---|
> |LLaMa|30.9|60.9|
> |LLaMa2|37.8|63.2|
>
> **[Q3] SMILES edit tasks.**
>
> Thank you for the suggestion. We conduct additional experiments on the forward reaction prediction task to show LLaMo’s capability in editing SMILES as suggested. We use the forward reaction prediction dataset from Mol-Instructions. The results (below table) indicate that our LLaMo consistently outperforms other models in all metrics.
>
> |Model|BLEU&uarr;|Levenshtein dist.&darr;|Tanimoto Sim.&uarr;|
> |---|---|---|---|
> |Galactica|46.8|35.02|0.16|
> |Text+Chem T5|78.2|20.41|0.71|
> |Mol-Instructions|65.4|27.26|0.31|
> |LLaMo|82.4|6.81|0.76|

---

> > ### Comment · Reviewer_rvcw · 2024-08-09
> >
> > Thanks for your detailed responses. It addresses all my problems. I raise my score from 6 to 7.

---

> ### Author Response · Authors · 2024-08-11
>
> We sincerely appreciate the reviewer's positive comments and increased rating.
>
> As the reviewer suggested, we will include a discussion on over-smoothing, the quality of GPT-generated data, and other tasks in the final version.

---

### Official Review · Reviewer_P4Gm · 2024-07-12

**Soundness:** 4
**Presentation:** 3
**Contribution:** 3
**Rating:** 7
**Confidence:** 5

**Summary:**

This paper presents LLaMo, a novel enabling LLMs and instruction tuning in molecular domain. To bridge the gap between different modalities, the paper also proposes a projector that transforms the graph representations into graph tokens level by level. The authors conduct extensive experiments and compare LLaMo with several proper baselines. The results are convincing.

**Strengths:**

1. The proposed method makes a good contribution to enable LLMs into the molecular domain. It successfully bridges the gap between language and graph modalities.
2. In terms of quality, the experiment setup is sound. Also, the authors conduct detailed and extensive experiments. It seems evident that LLaMo performs better than the baselines mentioned in the paper.
3. The paper is well-written and well-structured overall.

**Weaknesses:**

1. Though the authors conduct several experiments, they still lack some comparison between different GNN and LLM backbones. In section 5.3(Impact of Multi-level Graph Projector), will different GNN backbones harm or improve the results? Also, under the same experiment settings, the paper lacks the comparison between different LLM backbones. Such as the settings in Table 2, will the results be better for LLama-7b?
2. With several layers of GNNs, the efficiency of this projector may not be good.

**Questions:**

1. For line 138-139, could you explain in detail for the $\[p^{(l)}_1, .... , p^{(l)}_b\]$, such as what is the learnable prompts and how do you initialized it?
2. For the few-shot prompts part, how do you choose the few-shot examples?

**Limitations:**

The authors address the limitations of this work.

---

> ### Author Rebuttal · Authors · 2024-08-07
>
> **[W1] More performance comparison based on different GNN and LLM backbones.**
>
> Good question. We conduct additional experiments to evaluate performance based on different GNN and LLM backbones. Specifically, we compare pretrained Graph Convolutional Networks (GCNs)  with our base GNN backbone (GIN) and use LLaMa-7B as an alternative LLM backbone, as suggested. We use the same experimental setting for reporting Table 2. The experimental results for molecule description generation are reported in the table below.
>
> - **Different base GNN (GCN v.s. GIN (ours))**
>
>     The table demonstrates that the pre-trained GIN shows better performance than the pre-trained GCN, with a 4.8 improvement in the BLEU metric for the molecule description generation task. This result shows the superior expressivity of the GIN model, which aligns with the result in Table 1 of [1].
>
> - **Different base LLM (LLaMa-7B v.s. LLaMa2-7B (ours))**
>
>     The table shows that the LLaMa2-7B achieves a 6.9 performance improvement in the BLEU metric compared to LLaMa-7B. This indicates that a more powerful LLM enhances the performance of our LLaMo.
>
>
> | GNN | LLM | BLEU | METEOR |
> | --- | --- | --- | --- |
> | GCN | LLaMa2-7B | 33.0 | 61.8 |
> | GIN | LLaMa-7B | 30.9 | 60.9 |
> | GIN | LLaMa2-7B | 37.8 | 63.2 |
>
> **[W2] Efficiency of multi-level graph projector.**
>
> Good point. Since we do not increase the number of graph tokens and GNN layers, our multi-level graph projector maintains efficiency comparable to other projectors. To showcase it, we measure the inference time (second per generated token) of the model with various projectors, including our multi-level graph projector in the table below. From the table, our MGProj has similar efficiency to other projectors. Interestingly, the inference time difference between the setups without graphs and with graphs is minimal. We think that this is because the number of input text tokens is 108.87 in average, which is significantly larger than the number of graph tokens (32).
>
> | Projector | Time (sec/generated token) |
> | --- | --- |
> | w/o Graph | 0.056 |
> | MLP (w/ low-level) | 0.059 |
> | MLP (w/ high-level) | 0.059 |
> | MLP (w/ concat) | 0.059 |
> | Resampler | 0.058 |
> | MGProj (w/o motif) | 0.060 |
> | MGProj (ours) | 0.060 |
>
> **[Q1] Details of learnable prompts (L138-139).**
>
> The learnable prompts means learnable tokens $\mathbf{P}^{(l)} = \left[\mathbf{p}_1^{(l)}, \dots, \mathbf{p}_b^{(l)} \right]$ in L138. We initialize them with values drawn from the normal distribution.
>
> **[Q2] Details of choosing the few-shot examples.**
>
> As mentioned in the Section F of the supplement (L885-890), we select the exemplars from the train split of each dataset, based on their similarity to the target molecule. To calculate the similarity, we use the Tanimoto similarity metric, a widely-used metric for comparing molecular structures. We select the four molecules with the highest similarity scores to the target molecule.
>
> [1] Hu, Weihua, et al. "Strategies for pre-training graph neural networks." ICLR 2020.

---

> > ### Comment · Reviewer_P4Gm · 2024-08-13
> > **Official Comment by Reviewer P4Gm**
> >
> > Thank you for the clarification. It addresses my concerns, so I will raise my score to 7.

---

### Author Rebuttal · Authors · 2024-08-07

We appreciate all the reviewers for their time and efforts in reviewing our paper and insightful comments and questions. We are encouraged that the reviewers recognize multiple strengths in our paper, including:

- **Clear and effective design** that enables LLMs to operate within the molecular domain (P4Gm, rvcw, W78D) by successfully bridging the gap between language and graph modalities with the multi-level graph projector (P4Gm, 3eLt).
- **Extensive and detailed experiments** (P4Gm, rvcw, 3eLt) that demonstrate the effectiveness of the proposed model and the contribution of each component (rvcw).
- **A lot of qualitative examples** (3eLt), including illustrations of attention distribution that highlight the impact of the multi-level graph projector (W78D, 3eLt).
- **Sound and detailed experimental setups** (P4Gm) with code implementation (W78D).
- **Well-structured and clear writing** (P4Gm, rvcw, W78D), including comprehensive, step-by-step explanations of each component (3eLt).

We have tried our best to address the reviewers’ questions and concerns within the available time. We believe that incorporating this constructive feedback significantly enhances the quality of the paper. We sincerely thank all the reviewers for their valuable contributions. Please find individual responses to the comments below.

---

### Decision · Program_Chairs · 2024-09-25

**Decision:**

Accept (poster)

**Comment:**

The paper proposes a multi-level graph projector to create tokens which can complement LLM instructions. The multi-level nature is obtained by applying the embeddings from different GNN layers. It integrates well domain knowledge (e.g., different molecule modalities) and novel technology, contains an extensive evaluation, and shows good performance using state-of-the art LLMs. The paper also proposes a novel molecule + text dataset.

All four reviewers recommend to accept the paper and I recommend that as well.